# Implementation of the new integrated algorithm for diagnosis of drug-resistant tuberculosis in Karnataka State, India: How well are we doing?

Uma Shankar S. [1]*, Ajay M. V. Kumar[2,3,4], Nikhil Srinivasapura Venkateshmurthy[5,6], Divya Nair[7], Reena Kingsbury[1], Padmesha R.[1], Magesh Velu[1], Suganthi P.[1], Joydev Gupta[1], Jameel Ahmed[1], Puttaswamy G.[1], Somashekarayya Hiremath[1], Ravi K. Jaiswal[1], Rony Jose Kokkad[1], Somashekar N.[1]

1 National Tuberculosis Institute, Bangalore, India, 2 International Union Against Tuberculosis and Lung Disease (The Union), Paris, France, 3 International Union Against Tuberculosis and Lung Disease (The Union), South-East Asia Office, New Delhi, India, 4 Yenepoya Medical College, Yenepoya (Deemed to be University), Mangaluru, India, 5 Public Health Foundation of India, Gurgaon, India, 6 Harvard TH Chan School of Public Health, Boston, MA, United States of America, 7 The INCLEN Trust International, New Delhi, India

* nti@ntiindia.org.in, ushankars.2017@gmail.com

**Data Availability Statement:** All relevant data are within the manuscript and the tables and figures. The raw data is the routine programme data from

## Abstract

### Background

As per national policy, all diagnosed tuberculosis patients in India are to be tested using Xpert® MTB/RIF assay at the district level to diagnose rifampicin resistance. Regardless of the result, samples are transported to the reference laboratories for further testing: first-line Line Probe Assay (FL-LPA) for rifampicin-sensitive samples and second-line LPA(SL-LPA) for rifampicin-resistant samples. Based on the results, samples undergo culture and pheno-typic drug susceptibility testing. We assessed among patients diagnosed with tuberculosis at 13 selected Xpert laboratories of Karnataka state, India, i) the proportion whose samples reached the reference laboratories and among them, proportion who completed the diag-nostic algorithm ii) factors associated with non-reaching and non-completion and iii) the delays involved.

### Methods

This was a cohort study involving review of programme records. For each TB patient diag-nosed between 1st July and 31st August 2018 at the Xpert laboratory, we tracked the labora-tory register at the linked reference laboratory until 30th September (censor date) using Nikshay ID (a unique patient identifier), phone number, name, age and sex.

### Results

Of 1660 TB patients, 1208(73%) samples reached the reference laboratories and among those reached, 1124(93%) completed the algorithm. Of 1590 rifampicin-sensitive samples,

National TB Elimination Programme (NTEP), Ministry of Health & Family Welfare, Government of India. The sharing such programme data requires permission of the Deputy Director General of NTEP.

**Funding:** US, the corresponding author received the fund. The training program, within which this paper was developed, was funded by the Department for International Development (DFID), UK. In addition, funding for field data collection was provided by National Tuberculosis Institute, Bengaluru. The funders had no role in study design, data collection and analysis, decision to publish, or preparation of the manuscript.

**Competing interests:** It is declared that none of the authors have any competing interests.

1170(74%) reached and 1104(94%) completed the algorithm. Of 64 rifampicin-resistant samples, only 35(55%) reached and 17(49%) completed the algorithm. Samples from rifampicin-resistant TB, extra-pulmonary TB and two districts were less likely to reach the reference laboratory. Non-completion was more likely among rifampicin-resistant TB and sputum-negative samples. The median time for conducting and reporting results of Xpert® MTB/RIF was one day, of FL-LPA 5 days and of SL-LPA16 days.

## Conclusion

These findings are encouraging given the complexity of the algorithm. High non-reaching and non-completion rates in rifampicin-resistant patients is a major concern. Future research should focus on understanding the reasons for the gaps identified using qualitative research methods.

## Introduction

Tuberculosis (TB), an ancient disease whose cause was discovered nearly two centuries ago, still kills more people in the world, than any other infectious disease. TB has received unprecedented global attention in recent times, rightly so, and the first-ever United Nations high-level meeting on TB was held in September 2018 [1]. As a global community, we have pledged to end TB globally by 2030, with India committing to achieve this goal by 2025 [2, 3]. This may not be possible unless we achieve universal access to prevention, diagnosis and treatment of all forms of TB including drug-resistant tuberculosis (DR-TB).

DR-TB has emerged as a major public health crisis globally as well as in India. According to the World Health Organization (WHO), in 2018 an estimated 130,000 people in India developed rifampicin-resistant TB (RR-TB) or multidrug-resistant TB (MDR-TB) (defined as resistance to at-least isoniazid (H) and rifampicin (R), the two most effective first-line drugs). This accounts for one-fourth of the global burden of MDR/RR-TB [4]. The national drug resistance survey conducted in India during 2014–16 showed that the overall prevalence of MDR-TB was 6.2% among all patients (2.8% among new and 11.6% among previously treated patients). Among MDR-TB patients, 25% had pre-extensively drug-resistant TB. i.e., resistance to either fluoroquinolone (FQ) or second-line injectables (SLI), but not both and 1.3% had extensively drug-resistant TB (XDR-TB), defined as additional resistance to both FQ and SLI [5].

India initiated the programmatic management of DR-TB in 2007 [6]. During the initial years, diagnosis of DR-TB was based on the time-consuming, culture and phenotypic drug susceptibility testing (CDST) methods. There has been great progress over the years with the scale-up of WHO-approved rapid molecular diagnostic technologies such as Xpert® MTB/RIF assay and Line Probe Assay (LPA) [7, 8]. Despite this, only 29% of all estimated RR/MDR-TB patients in 2017 were reported to be diagnosed in India [9]. However, of the patients diagnosed, it was encouraging to note that 92% were started on treatment. Similar findings were also reported in a systematic review and meta-analysis conducted on TB cascade of care in India [10]. These indicate relatively larger gaps in the diagnosis of DR-TB rather than treatment initiation. Such undiagnosed cases are likely to spread the disease and continue the chain of transmission in the community.

Another cause of concern has been treatment success, which remains poor at 46% for MDR-TB patients and 28% among XDR-TB patients in India [11]. The possible reasons for this might be related to delays in diagnosis of additional drug resistance [10, 12–14] and non-

initiation of appropriate treatment based on the drug resistance patterns. This may lead to amplification of resistance and poorer treatment outcomes.

To address these concerns, several measures have been taken by the Government of India. Xpert[®] MTB/RIF assays were pilot-tested at 18 sites across India in 2012 and it was found that decentralized deployment was feasible [15]. Accordingly, Xpert labs were scaled up across the country and are now available at district and sub-district levels, while LPA and CDST services are available at state level (provincial) and sub-provincial reference laboratories. A policy of universal drug susceptibility testing (UDST) was introduced in 2017, wherein all diagnosed TB patients, whether microbiologically confirmed or clinically diagnosed are to be routinely offered Xpert[®] MTB/RIF assay to assess rifampicin resistance [16]. If MTB is detected, the samples are transported to the reference laboratories for further testing by LPA and liquid culture and DST methods (Fig 1).

Since the launch of this new integrated diagnostic algorithm for DR-TB in India, there has not been any systematic assessment as to how well this is being implemented. Hence, we aimed to assess, among all the TB patients diagnosed at selected Xpert laboratories of Karnataka state, India, i) the proportion whose samples reached the reference laboratories and among them, the proportion who completed the diagnostic algorithm ii) demographic and clinical factors associated with non-reaching and non-completion of algorithm and iii) the delays involved at each step of the diagnostic cascade.

## Methods

### Study design

This was a cohort study involving analysis of routine programmatic data extracted from the laboratory registers maintained at the selected Xpert laboratories and their linked reference laboratories.

### Setting

**General setting.** The study was conducted in Karnataka, one of the large states in South India. The state has a population of 61 million with about 60% living in rural areas [17]. The sex ratio is 979 females per 1,000 males and the literacy rates are 85% among men and 72% among women [17]. In Karnataka, the prevalence of TB is estimated to be 180 persons per 100,000 [17].

**Specific setting.** The general healthcare services are delivered through a network of government and private health facilities at primary, secondary and tertiary level. The Revised National Tuberculosis Programme (RNTCP) is implemented by the Karnataka State through 31 district TB centres, 188 sub-district level TB units and 688 sputum microscopy centres functioning under a system of quality assurance [18]. There are a total of 66 (65 permanent and 1 mobile) Xpert laboratories in the state with at-least one laboratory in each district. These have been linked to one of the four designated reference laboratories which include one National Reference Laboratory housed in National Tuberculosis Institute (NTI) and one Intermediate Reference Laboratory (IRL), both located in Bengaluru; and two other laboratories located at Hubli and Raichur. These laboratories are equipped and proficient to conduct first-line LPA (FL-LPA using GenoType MTBDR*plus* VER 2.0), second-line LPA (SL-LPA using GenoType MTBDR*sl* VER 2.0, Hain Lifescience GmbH) and liquid CDST using Mycobacterial Growth Indicator Tube 960 system (BD MGIT™). The samples are transported through the human carriers or more routinely, a courier agency. Human carriers are the employees of the general health system (multipurpose health workers, etc.) or programme staff (TB- Health Visitors) stationed at the facility. During the study period, a Non-Governmental Organization (NGO)

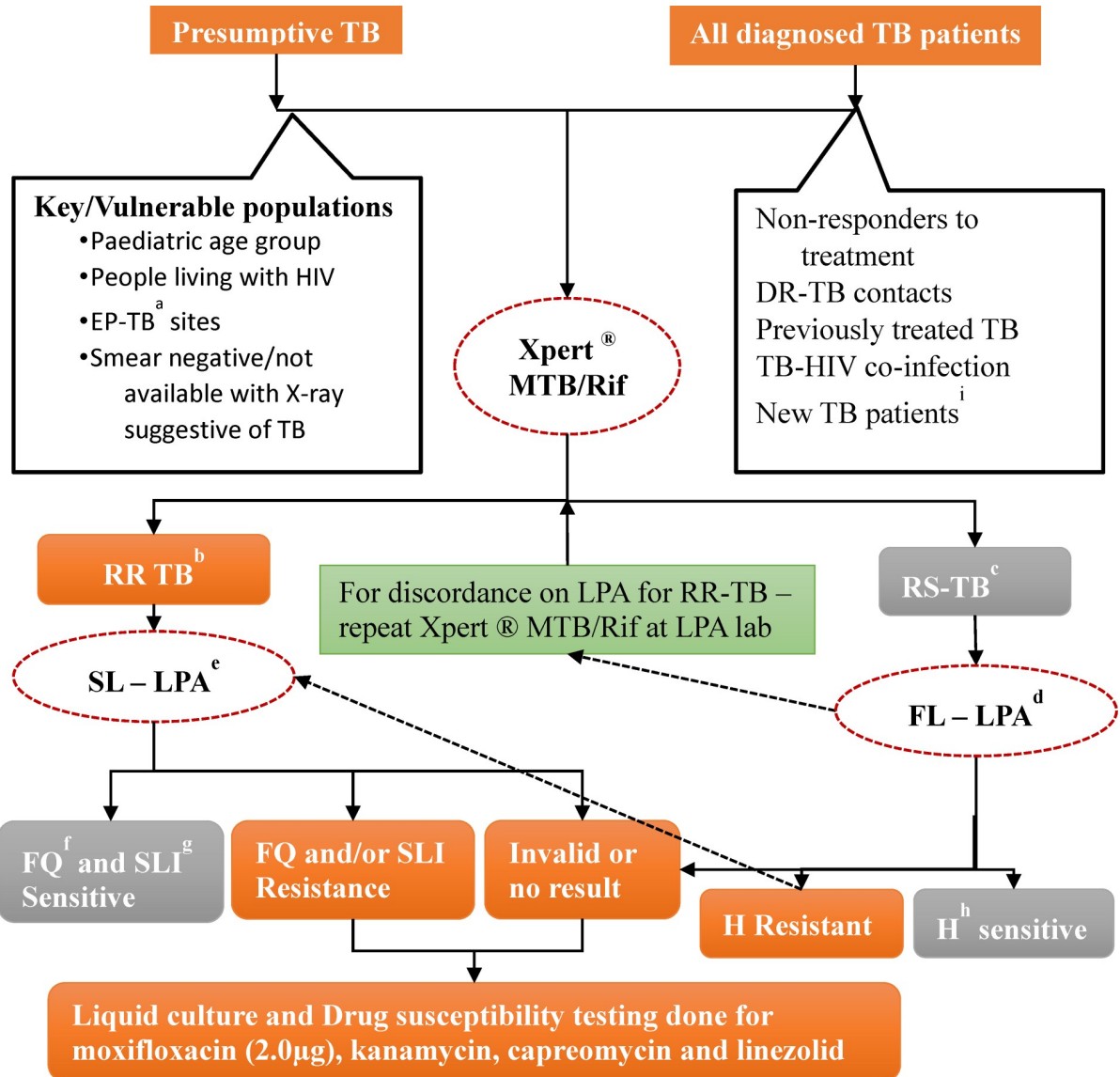

**Fig 1. The integrated drug resistant TB diagnostic algorithm as per the Programmatic Management of Drug Resistant TB Guidelines, Revised National TB Programme, India, 2017.** [a]Extra-pulmonary TB [b]Rifampicin resistant TB [c]Rifampicin sensitive TB [d]Fluoroquinolone [e]Second-line injectables [f]First-line Line Probe Assay [g]Second-line Line Probe Assay [h]Isoniazid [i]All diagnosed TB patients (clinically diagnosed or microbiologically confirmed) were offered Xpert® MTB/Rif assay to know the rifampicin sensitivity status as per the Universal Drug Susceptibility policy since April 2018 in Karnataka state, India.

with external funding was assisting in transportation of the samples between the laboratories in districts around Bengaluru city by deploying human carriers.

## The integrated diagnostic algorithm

Karnataka adopted the policy of UDST in April 2018. As per the new algorithm, all the TB patients are offered Xpert® MTB/RIF assay. It is expected that two sputum samples are collected from the patients at the peripheral health institutions (including sputum microscopy centres) in conical tubes and transported in triple layered package in cool chain (temperature maintained at 8°C to 20°C) to the Xpert laboratories situated at district or sub-district level.

One of the samples is used for Xpert testing and for all those diagnosed with TB, the other sample is transported to the linked reference laboratories.

Further testing at the reference laboratory depends on the Xpert® MTB/RIF assay results, which are expected to be available in one day. If rifampicin resistance is detected in a new patient, the second sample is used at the Xpert laboratory for a repeat test to confirm rifampicin resistance. If confirmed, the patients are requested to provide fresh sputum specimens to send to reference laboratories. In a previously treated patient, no such confirmatory testing is required, and the second sample is transported to the reference laboratory for SL-LPA to identify mutations in the MTb genes which could confer additional resistance to FQ and/or SLI class of drugs. In case any test was repeated more than once for any reason, then only the final test results were recorded. Awaiting results of SL-LPA, the patient is started on second-line treatment using the short course MDR-TB regimen. Generally, LPA results are expected within 72 hours. If SL-LPA shows no evidence of additional resistance, treatment with short course MDR-TB regimen is continued. If additional resistance is detected, Pre-XDR or XDR TB regimens are started based on resistance to either or both FQ and SLI and further testing is done using liquid CDST to test for resistance to individual or additional drugs (moxifloxacin, linezolid, kanamycin and capreomycin) and based on the final resistance pattern, an individualised DST-based treatment regimen is started.

If Xpert® MTB/RIF assay results show rifampicin sensitive TB, the patient is started on first line treatment while their second sample is sent for FL-LPA. If found sensitive to isoniazid, the patient is continued on first-line treatment; if resistant to isoniazid, regimen for isoniazid resistant TB (wherein resistance to the other first-line anti-TB drugs is not known) is initiated and SL-LPA test is conducted; further management is as described above (**Fig 1**).

If LPA is not able to provide a valid result or in case of smear-negative samples, the samples are cultured, and LPA is run on the culture isolate. When sputum sample is not of adequate quality or insufficient quantity to obtain a culture result, patients are contacted again for fresh samples. The laboratory turnaround times for Xpert® MTB/RIF is 2 hours, for both the LPAs is 72 hours and for liquid CDST is 42 days.

## Study population and study period

All the TB patients diagnosed using Xpert® MTB/RIF assay at the 13 Xpert laboratories situated in ten selected districts between 1st July and 31st August 2018 were included. The districts were selected based on convenience and feasibility of data collection and all the Xpert laboratories within the selected districts are included. The districts were Bengaluru city, Bengaluru rural, Bengaluru urban, Chamarajanagar, Kodagu, Kolar, Mandya, Mysore, Ramanagara and Shivamogga. While the first three districts were linked to NTI, the rest were linked to IRL Bangalore, both located in South Karnataka. The data was extracted from laboratory registers between February and April 2019.

## Data variables, sources of data and data collection

All the variables required were extracted from the laboratory registers maintained at the Xpert and the reference laboratories. These included dates of specimen collection, transportation, reporting the test results; type of sample, details of resistance detected and key population (which included high risk groups for TB like tobacco users, people living with HIV (PLHIV), diabetes etc.). To facilitate efficient tracking of patients between the registers, digitization of both the registers was done in quality-assured manner (double entry and validation was done for the data extracted from laboratory registers kept at Xpert laboratories) using EpiData software (v3.1, EpiData Association, Odense, Denmark) by trained data entry operators. We used

password-protected electronic data capture systems which were installed on secure desktop systems with restricted access to study investigators only. In case multiple samples from the same patient were tested, the result of the latest sample was captured and used for analysis.

We first digitized the Xpert laboratory registers for the period 1$^{st}$ July to 31$^{st}$ August 2018 (database 1). We then digitized the laboratory register of the reference laboratories for the period 1$^{st}$ July to 30$^{th}$ September 2018 (database 2) to ensure that every patient found to be Mtb detected at the Xpert laboratory is tracked for a period of at-least one month in the reference laboratory register.

For each patient listed in database 1, we searched the database 2 to assess if the patient has reached the reference laboratory using Nikshay ID (a unique patient identification number generated on a case-based web-based TB notification and surveillance platform) as the primary tracking variable. If we did not find a match using Nikshay ID, we then used the mobile phone number. If we failed to get a match using mobile phone number, we used the combination of patient's name, age and sex. If we did not find a match using any of the above variables, we considered that the sample had not reached the reference laboratory. A master database thus created was de-identified and used for analysis.

## Analysis and statistics

Data analysis was done using EpiData (v 2.2.2.182) and STATA (v 12.1, Statacorp, Texas, USA) software. The key outcome variables were: i) proportion of TB patients diagnosed at Xpert laboratories whose samples did not reach the reference laboratory for further testing and among the samples reached, ii) proportion who did not complete the diagnostic algorithm. The median duration and interquartile range (IQR) between receipt of specimens and reporting of results at the two laboratories were calculated. The operational definitions of the outcome variables are described in **Table 1**.

Association between the outcomes (non-reaching and non-completion of diagnostic algorithm) and various demographic and clinical factors were examined using chi-square test and measured using unadjusted relative risk (RR) with 95% confidence intervals (CI). We used poisson regression to calculate adjusted relative risks and 95% CI. We used an exploratory approach and included all the variables in adjusted analysis in the multivariable model. A p-value of ≤0.05 were considered statistically significant.

## Ethics

Ethics approval was obtained from the Institutional Ethics Committee of the NTI, Bangalore, India, and the Ethics Advisory Group of the International Union Against Tuberculosis and Lung Disease, Paris, France (Number 128/18). Since the study involved review of existing programme records without any direct interaction with human participants, the need for individual informed consent was exempted by the ethics committees. Permission to conduct the study was obtained from the director of NTI and the State TB Officer of Karnataka State.

## Results

### Demographic and clinical profile

There were a total of 1660 TB patients tested by Xpert$^{®}$ MTB/Rif during the study period. Of them, 1199 (72%) were males and the mean (SD) age was 42 (16) years. Most of the patients (1431, 86%) had been referred from the government health facilities for Xpert MTB/Rif testing and sputum was most common specimen tested (1449, 87%). Those who did not belong to any key population group were 928 (56%) and among the key population 218 (13.1%) were tobacco

**Table 1. Operational definitions of the outcome variables used to assess the reach, completion, dates of collection, receipt and reporting of results for various diagnostic tests used for diagnosis of DR-TB in the integrated DR-TB diagnostic algorithm.**

| Indicator | Operational Definition |
|---|---|
| Reaching the reference laboratory | A patient found to be Mtb detected in Xpert laboratory, who was documented to have received services in the reference laboratory was considered as having reached. If a match was found by Nikshay ID[a], mobile phone number or name-age-sex, then it was considered that the patient's sample had reached the reference laboratory. |
| Patients completing the diagnostic algorithm | This is a composite indicator and was calculated among patients whose samples reached the reference laboratory. Patients fulfilling the following criteria was considered as having completed the diagnostic algorithm.<br>• Rifampicin sensitive on Xpert® MTB/Rif and Isoniazid and Rifampicin sensitive on FL-LPA[b] at reference laboratory<br>• Rifampicin sensitive on Xpert® MTB/Rif and resistance to Isoniazid and/or Rifampicin on FL-LPA and sensitive to second-line drugs on SL-LPA[c]<br>• Rifampicin sensitive on Xpert® MTB/Rif and resistance to Isoniazid and/or Rifampicin on FL-LPA and resistant on SL-LPA and culture done<br>• Rifampicin resistant on Xpert® MTB/Rif and sensitive on SL-LPA<br>• Rifampicin resistant on Xpert® MTB/Rif and resistant on SL-LPA and culture done<br>• Culture done on samples with indeterminate or no results on either FL-LPA or SL-LPA |
| Date of specimen collection | This is the date of specimen collection as documented in the laboratory register maintained at Xpert laboratory. |
| Date of specimen receipt | This is the date of specimen receipt as documented in the laboratory register maintained at Xpert laboratory or reference laboratory. |
| Date of reporting results | This is the date of reporting results as documented in the laboratory register at the Xpert laboratory or reference laboratory for a diagnostic technology. |

[a] Unique patient identifier generated on the case-based web-based TB notification platform [b] First line—Line Probe Assay [c] Second line–Line Probe Assay

users and 208 (13%) were PLHIV. Among those documented, 1135 (83%) were newly diagnosed cases. Of the total TB patients, 1590 (96%) were rifampicin sensitive, 64 (4%) were rifampicin resistant, four patients had indeterminate results and two did not have any information on the rifampicin resistance status. (**Table 2**)

## Samples reaching and completing the diagnostic algorithm at the reference laboratories

Out of the total 1660 TB patients tested by Xpert® MTB/Rif six were not included in the analysis for samples reaching and completing the diagnostic algorithm (R resistance reports were not available for 2 and were indeterminate for 4). Of the 1590 rifampicin-sensitive patient samples detected at Xpert laboratory, 1170 (74%) reached the reference laboratory and all were eligible for testing with FL-LPA. Among them, FL-LPA was done for 1131 (97%) and 51 (5%) were diagnosed to have isoniazid resistance and three had rifampicin resistance. Thus, among the 54 eligible samples (with resistance to isoniazid or rifampicin), SL-LPA was done for 11 (20%). Culture was done on 41/107 (38%) samples whenever either FL-LPA or SL-LPA was not done or did not yield a result. Thus, a total of 1104 (94%) of rifampicin-sensitive TB patients whose samples were successfully received at the reference laboratory were considered to have completed the diagnostic algorithm (**Fig 2**).

Of the 64 rifampicin-resistant patient samples detected at Xpert laboratory, 35 (55%) reached the reference laboratory and all were eligible for testing with SL-LPA. Among them, SL-LPA was done for 14 (40%) and nine samples were sensitive to FQ and SLI. Culture was

**Table 2. Demographic and clinical profile of TB patients diagnosed by Xpert® MTB/Rif assay between 1st July and 31st August 2018 at t13 selected Xpert laboratories in Karnataka, India.**

| Variable | Number | (%) |
|---|---:|---|
| **Total** | 1660 | (100) |
| **Age (years)** | | |
| ≤ 14 | 20 | (1.2) |
| 15–29 | 396 | (23.9) |
| 30–44 | 523 | (31.5) |
| 45–59 | 429 | (25.8) |
| ≥ 60 | 284 | (17.1) |
| Not recorded | 8 | (0.5) |
| **Sex** | | |
| Male | 1199 | (72.2) |
| Female | 457 | (27.5) |
| Not recorded | 4 | (0.2) |
| **Key population** | | |
| Contact of TB patient | 148 | (8.9) |
| Tobacco user | 218 | (13.1) |
| Urban slum dweller | 135 | (8.1) |
| People living with HIV | 208 | (12.5) |
| Others[a] | 23 | (1.4) |
| non-key population | 928 | (55.9) |
| **Type of referring health facility** | | |
| Government health facility | 1431 | (86.2) |
| Private health facility | 215 | (13.0) |
| Not recorded | 14 | (0.8) |
| **Type of TB patient** | | |
| New | 1135 | (68.4) |
| Previously treated | 236 | (14.2) |
| Not recorded | 289 | (17.4) |
| **Type of specimen** | | |
| Pulmonary | 1449 | (87.3) |
| Extra-pulmonary | 66 | (4.0) |
| Not recorded | 145 | (8.7) |
| **Specimen condition on receipt at Xpert lab** | | |
| Mucopurulent[b] | 1019 | (61.4) |
| Blood Stained[b] | 3 | (0.2) |
| Saliva[b] | 403 | (24.3) |
| Not recorded | 235 | (14.2) |
| **Rifampicin resistance results at Xpert lab** | | |
| Rifampicin Sensitive | 1590 | (95.8) |
| Rifampicin Resistant | 64 | (3.9) |
| Rifampicin Indeterminate | 4 | (0.2) |
| Not recorded | 2 | (0.1) |

[a] Others include people with diabetes, migrants, miners, prisoners etc.

[b] these conditions apply only for sputum samples.

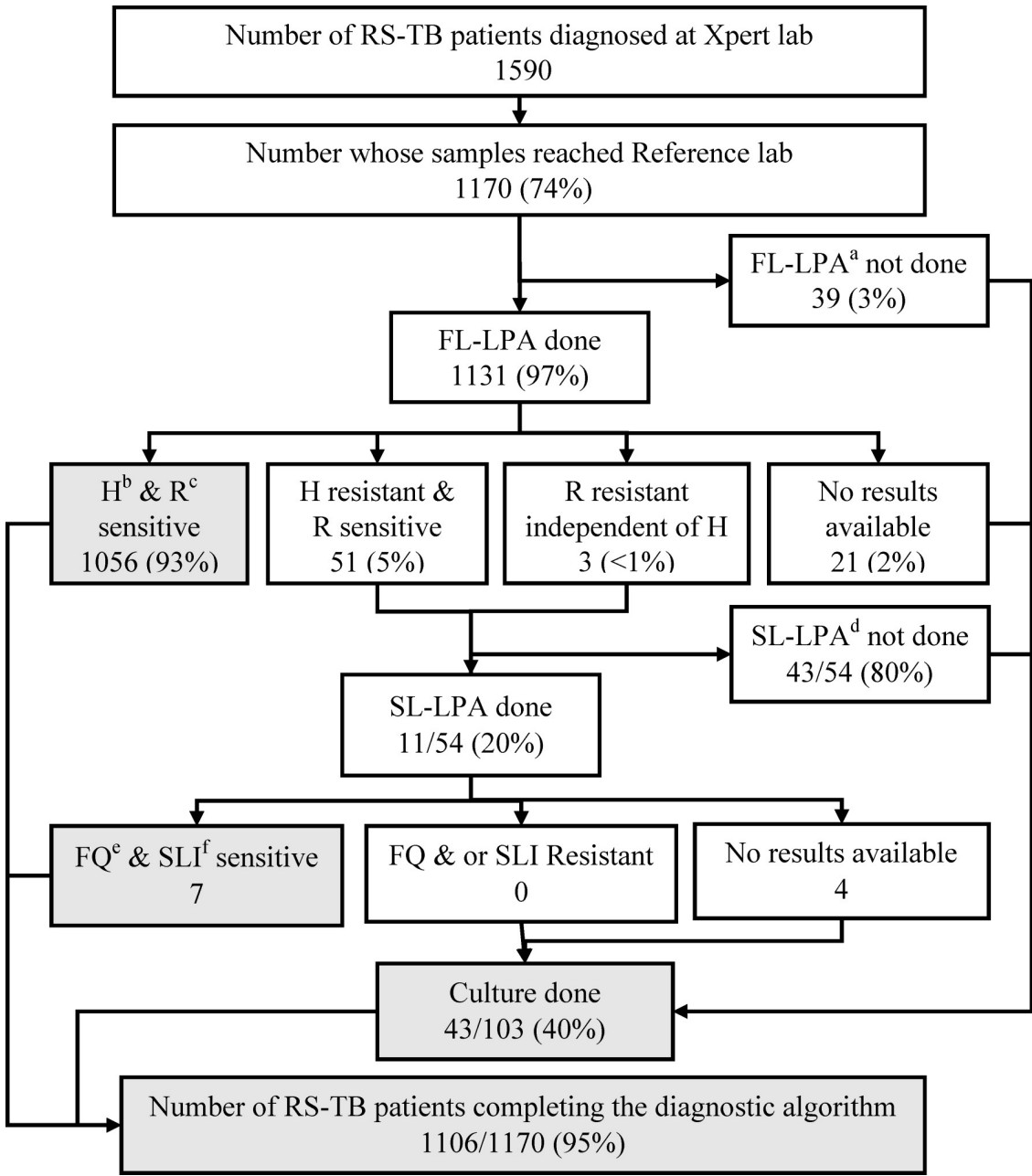

**Fig 2. Cascade of integrated DR-TB diagnostic algorithm for Rifampicin sensitive TB patients (RS-TB) diagnosed at the 13 selected Xpert laboratories between 1st July and 31st August 2018 in Karnataka, India.** [a] First line–Line Probe Assay [b] Isoniazid [c] Rifampicin [d] Two were H resistant and one was H sensitive [e] Second line–Line Probe Assay [f] Fluoroquinolone [g] Second line injectable.

done on eight samples where SL-LPA results were resistant or unknown. Thus, a total of 17 (49%) of rifampicin-resistant TB patients were considered to have completed the diagnostic algorithm (**Fig 3**).

Overall, out of the total 1660 samples, 452 (27%) samples did not reach the reference laboratories and among those reached, 84 (7%) did not complete the DR-TB diagnostic algorithm.

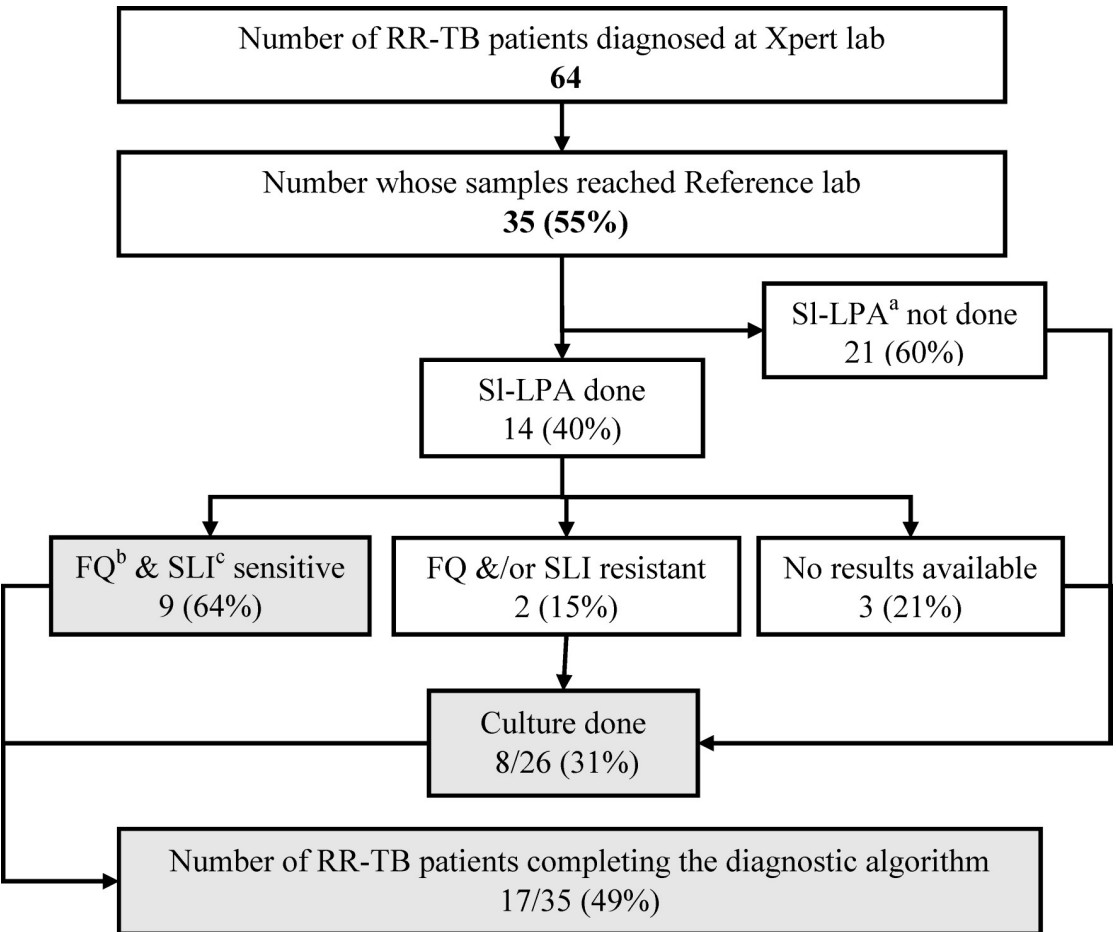

**Fig 3. Cascade of integrated DR-TB diagnostic algorithm for Rifampicin resistant TB (RR-TB) patients diagnosed at the 13 Xpert laboratories between 1st July and 31st August 2018 in Karnataka, India.** [a] Second line–Line Probe Assay [b] Fluoroquinolone [c] Second line injectable.

### Delays in the diagnostic pathway

Dates to calculate the delays were not consistently recorded in the laboratory registers. A total of 1250/1554 (80%) samples collected at the microscopy centres were transported within a day to the Xpert laboratory and the results were available within a median duration of one day. The median time duration between receipt of sample at the Xpert laboratory to receipt at the linked reference laboratory after transportation was 5 (IQR 3–7) days. On sample receipt at the reference laboratory, the median time taken for conducting and reporting the result of FL-LPA was 5 (IQR 4–6) days (among rifampicin-sensitive TB patients), of SL-LPA was 16 (IQR 8–30) days (among rifampicin-resistant TB patients) and 18 (IQR 11–37) days (among rifampicin-sensitive TB patients who underwent FL-LPA first and then SL-LPA). (**Table 3**)

### Factors associated with non-reach and non-completion

Factors associated with samples not reaching the reference laboratories are shown in **Table 4**. On adjusted analysis, we found that rifampicin-resistant TB samples, extra-pulmonary samples and samples from two districts (Chamarajnagara and Mysore) were less likely to reach the reference laboratory.

**Table 3. Timing of specimen collection, receipt and reporting of results of TB patients diagnosed between 1st July and 31st August 2018 at the 13 selected Xpert laboratories and their two linked reference laboratories in Karnataka, India.**

| Time durations between various laboratory steps | No. eligible | No. with valid dates | (%) | Median (IQR) |
|---|---|---|---|---|
| Specimen collection at DMC[a] and receipt at Xpert site | 1660 | 1554 | (93.6) | 0 (0–1) |
| Specimen receipt and reporting of results at Xpert site | 1660 | 1066 | (64.2) | 1 (0–2) |
| Specimen receipt at Xpert site and receipt at reference laboratory | 1205 | 1126 | (93.4) | 5 (3–7) |
| Specimen receipt at reference laboratory and FL-LPA[b] result reporting (for RS-TB[c] patients) | 1170 | 1116 | (95.4) | 5 (4–6) |
| Specimen receipt at reference laboratory and SL-LPA[d] result reporting (for RR-TB[e] patients) | 35 | 32 | (9.1) | 16 (8–30) |
| Specimen receipt at reference laboratory and SL-LPA reporting (for RS-TB patients whose FL-LPA result was resistant) | 54 | 10 | (18.5) | 18 (11–37) |

[a] Designated microscopy center

[b] First line–Line probe assay

[c] Rifampicin sensitive tuberculosis

[d] Second line–Line probe assay

[e] Rifampicin resistant tuberculosis

**Table 4. Factors associated with samples not reaching the linked reference laboratories from patients diagnosed with TB at the 13 selected Xpert laboratories between 1st July and 31st August 2018 in Karnataka, India.**

| Variable | Total | N | (%) | RR[b] | (95% CI) | aRR[d] | (95% CI[e]) |
|---|---|---|---|---|---|---|---|
| Total | 1660 | 452 | (27.2) | | | | |
| **Health Facility type** | | | | | | | |
| Government | 1431 | 377 | (26.3) | Ref[c] | | Ref | |
| Private | 215 | 70 | (32.6) | 1.24 | (1.00–1.53) | 1.04 | (0.84–1.28) |
| **Specimen type** | | | | | | | |
| Sputum | 1449 | 360 | (24.8) | Ref | | Ref | |
| Extra-Pulmonary | 66 | 40 | (60.6) | 2.45 | (1.97–3.02) | **2.52** | **(1.98–3.21)** |
| Not recorded | 145 | 52 | (35.9) | 1.44 | (1.14–1.83) | 1.58 | (0.89–2.81) |
| **District name** | | | | | | | |
| Chamarajanagar | 129 | 48 | (37.2) | 2.07 | (1.48–2.91) | **2.19** | **(1.57–3.08)** |
| Bengaluru city | 273 | 49 | (17.9) | Ref | | Ref | |
| Bengaluru rural | 120 | 22 | (18.3) | 1.02 | (0.65–1.61) | 0.94 | (0.59–1.51) |
| Bengaluru urban | 207 | 38 | (18.4) | 1.02 | (0.70–1.50) | 1.01 | (0.69–1.48) |
| Kodagu | 52 | 9 | (17.3) | 0.96 | (0.51–1.84) | 1.00 | (0.51–1.93) |
| Kolar | 143 | 48 | (33.6) | 1.87 | (1.32–2.63) | 1.30 | (0.69–2.43) |
| Mandya | 214 | 54 | (25.2) | 1.41 | (1.00–1.99) | 1.38 | (0.98–1.93) |
| Mysore | 224 | 129 | (57.6) | 3.21 | (2.43–4.23) | **3.31** | **(2.51–4.38)** |
| Ramanagara | 116 | 21 | (18.1) | 1.01 | (0.63–1.60) | 1.11 | (0.70–1.75) |
| Shivamogga | 182 | 34 | (18.7) | 1.04 | (0.70–1.55) | 1.07 | (0.73–1.57) |
| **R[a] resistance Results** | | | | | | | |
| R Sensitive | 1590 | 420 | (26.4) | Ref | | Ref | |
| R Resistant | 64 | 29 | (45.3) | 1.72 | (1.29–2.27) | **1.77** | **(1.32–2.37)** |

N = Samples not reaching the linked reference laboratories for further testing from the Xpert labs.

[a] Rifampicin

[b] Relative risk

[c] Reference

[d] Adjusted relative risk

[e] Confidence Interval

Factors with relative risk in bold font are statistically significant (p value <0.05)

**Table 5. Factors associated with non-completion of the DR-TB diagnostic algorithm among the samples reaching the two linked reference labs from those diagnosed with TB at the 13 selected Xpert laboratories between 1st July and 31st August 2018 in Karnataka, India.**

| Variable | Total | N | (%) | RR[d] | (95% CI) | aRR[f] | (95% CI[g]) |
|---|---|---|---|---|---|---|---|
| Total records | 1208 | 84 | (7.0) | | | | |
| **Health Facility type** | | | | | | | |
| Government | 1054 | 68 | (6.5) | Ref[e] | | Ref | |
| Private | 145 | 14 | (9.7) | 1.50 | (0.86–2.59) | 1.64 | (0.95–2.82) |
| **Specimen type** | | | | | | | |
| Sputum | 1089 | 80 | (7.3) | Ref | | Ref | |
| Extra-Pulmonary | 26 | 1 | (3.8) | 0.52 | (0.08–3.62) | 0.53 | (0.07–4.04) |
| Not recorded | 93 | 3 | (3.2) | 0.44 | (0.14–1.36) | 0.35 | (0.09–1.42) |
| **R[a] resistance Results** | | | | | | | |
| R[a] Sensitive | 1170 | 64 | (5.5) | Ref | | Ref | |
| R[a] Resistant | 35 | 17 | (48.6) | 8.88 | (5.86–13.46) | **8.47** | **(5.36–13.39)** |
| **Reference laboratory (RL)** | | | | | | | |
| NTI[b] Bengaluru | 491 | 38 | (7.7) | 1.21 | (0.80–1.83) | 0.68 | (0.42–1.12) |
| IRL[c] Bengaluru | 717 | 46 | (6.4) | Ref | | Ref | |
| **Smear microscopy results at RL** | | | | | | | |
| Negative | 100 | 16 | (16.0) | 2.67 | (1.61–4.44) | **2.69** | **(1.42–5.12)** |
| Positive | 1069 | 64 | (6.0) | Ref | | Ref | |
| Not recorded | 39 | 4 | (10.3) | 1.71 | (0.66–4.47) | 1.45 | (0.48–4.43) |

N = non-completion of the DR-TB diagnostic algorithm among the specimens reaching the two linked reference labs from the Xpert labs

Not recorded- Not recorded were excluded from the model for age group, sex, health facility type, R resistance results and smear microscopy results at RL.

[a] Rifampicin

[b] National Tuberculosis Institute

[c] Intermediate reference laboratory

[d] Relative risk

[e] Reference

[f] Adjusted relative risk

[g] Confidence interval

Factors with relative risk in bold font are statistically significant (p value <0.05)

Factors associated with non-completion of the DR-TB diagnostic algorithm are shown in **Table 5**. On adjusted analysis, we found that rifampicin-resistant TB samples, and sputum microscopy negative samples were less likely to complete the diagnostic algorithm. Patient level variables like age, sex, key population and type of patient were not included in the Tables 4 and 5, considering these variables did not have any effect on the outcomes. Statistical output sheets for Tables 4 and 5 are provided as S1 File.

## Discussion

This is the first study from India to assess the gaps and delays in implementation of an integrated DR-TB diagnostic algorithm under programme settings. We found that about three-fourths of the samples reached the reference laboratory and of them, more than 90% completed the diagnostic algorithm. No gross delays were observed in the diagnostic cascade except for the pathway involving SL-LPA testing. These findings are encouraging given the complexity of the algorithm, involving several tests to be administered in a specific sequence, involving transportation of samples between the Xpert and reference laboratories located far away from each other (distance ranging from 1 to 280 kilometres).

However, there were some gaps too. Approximately a quarter of the samples did not reach the reference laboratories. This was more likely with extra-pulmonary samples, RR-TB samples and samples from some districts. The challenges associated with obtaining adequate sample volume, transportation and processing of the extra-pulmonary samples have been well documented in previous studies [19–21]. In one of the earlier studies for diagnosis of MDR-TB, patients with extra pulmonary TB had 50% higher risk of not getting tested when compared to patient with pulmonary TB [22]. This might be due to inadequate capacity of people involved in the collection of extra-pulmonary specimens, the methods and volume/size of the sample required, non-availability of mechanisms for early transportation of these samples and non-availability of concentration methods or testing capacity for such samples at all the Xpert laboratories.

We were surprised by the gaps in the transport of RR-TB samples because they are generally accorded higher priority by the RNTCP. We speculate that this might be due to the practice of requesting for additional specimens from the patients after being diagnosed as RR-TB at Xpert laboratories especially for new cases (diagnosed with TB first time) and not receiving them in time or due to non-transportation of the samples from the Xpert laboratories. This needs further investigation.

The samples reaching the reference laboratory from Xpert laboratories situated in districts closer to Bangalore city was better. This may be due to the presence of system of human carriers for sample transport in these districts, who were deployed by a non-governmental organization with the support of external funding. A higher proportion of specimens from a couple of districts located farther away did not reach the reference laboratories. This calls for a review of training of key health personnel and sputum collection and transportation mechanisms in these districts.

Completion rates were excellent (95% among rifampicin-sensitive) once the samples reached the reference laboratories, except in sputum microscopy negative and RR-TB samples (49%). This may be explained by the fact that smear microscopy negative samples must be cultured before testing by LPA and if cultures did not yield growth, it required recollection of specimens from the patients–these might lead to non-completion of the diagnostic algorithm or delays in the process. The non-completion rates in RR-TB samples might be related to limitations of the first-generation SL-LPA technology with high rates of indeterminate or invalid results leading to non-availability of results, which could be as high as 44% when tested indirectly on culture isolates in smear microscopy negative samples as per the WHO policy guidance document on the usage of LPA [8]. As a result, the specimens needed to be re-tested multiple times before obtaining a valid result. This may explain why delays with SL-LPA were three times more when compared to FL-LPA, though valid dates of receipt and reporting of SL-LPA results were not available for all samples. This needs to be addressed on priority and requires further investigation A second-generation SL-LPA technology has been validated and recommended for usage by WHO, which may resolve these challenges [8].

## Strengths

First, we performed a rigorous assessment and took special efforts to ensure quality of data which included double entry and validation, wherever possible. Second, we had a large sample size which helped in performing a robust analysis and minimize the effect of random variation. Since this was an operational research done using the programme data, it reflects the field realities and actual field performance of different diagnostic tests in providing early and accurate diagnosis of DR-TB.

## Limitations

One of the major limitations was that the Nikshay ID was not documented for all TB patients. This made the tracking process challenging and we had to rely on using other variables such as mobile phone numbers, name, age and sex. As a result, we may have underestimated the proportion of samples reached. We have not examined for any interactions, given the fewer variables and low statistical power. We conducted the study in laboratories located in the selected districts situated in southern part of Karnataka state, thus limiting the generalizability of findings to other parts of the state.

## Programme implications

There are several implications of the study findings. First, we need to assess the reasons for gaps in the diagnostic cascade using qualitative research methods such as key informant interviews and a bottleneck analysis of the laboratory networks. This will inform specific measures that are required to address the issue. Second, comprehensive measures need to be put in place to improve specimen collection and transportation to reduce the delays involved. Third, the RNTCP needs to strengthen the documentation of Nikshay ID in all documents used along the diagnostic cascade, especially in the laboratory requisition forms filled at the time of sample collection which acts as a source document for updating the entries into the laboratory registers maintained at Xpert and reference laboratories. Steps must be taken to strengthen the real-time updating of results of FL-LPA, SL-LPA and CDST in laboratory registers and online in Nikshay for easy tracking and monitoring. This will also enable cohort-wise analysis and periodic review. Finally, we recommend repeating similar studies in other areas of the state and the country.

In conclusion, we found that approximately 75% of the samples of TB patients diagnosed in the Xpert laboratories of the southern part of Karnataka State reached the reference laboratory and of them, more than 90% completed the diagnostic algorithm. Some gaps were noted, wherein 25% of the specimens were not transported to the reference laboratories for further testing and non-completion of diagnostic algorithm, especially with respect to extra-pulmonary, sputum smear-negative and RR-TB samples. We appreciate the performance of the RNTCP in implementing such a complex diagnostic algorithm in an efficient manner, but there is scope for further improvement. Further research involving Xpert sites representing the whole state and also exploring the reasons for delays and non-completion is necessary to provide constructive feedback to programme managers for reducing the time and improving the efficiency in completion of the integrated DR-TB diagnostic algorithm.

## Supporting information

**S1 File.**
(DOCX)

## Acknowledgments

This research was conducted through the Structured Operational Research and Training Initiative (SORT IT), a global partnership led by the Special Programme for Research and Training in Tropical Diseases at the World Health Organization (WHO/TDR). The model is based on a course developed jointly by the International Union Against Tuberculosis and Lung Disease (The Union) and Medécins sans Frontières (MSF/Doctors Without Borders). The specific SORT IT programme which resulted in this publication was jointly developed and implemented by: The Union South-East Asia Office, New Delhi, India; the Centre for Operational

Research, The Union, Paris, France; Medécins sans Frontières (MSF/Doctors Without Borders), India; Department of Preventive and Social Medicine, Jawaharlal Institute of Postgraduate Medical Education and Research, Puducherry, India; Department of Community Medicine, All India Institute of Medical Sciences, Nagpur, India; Department of Community Medicine, ESIC Medical College and PGIMSR, Bengaluru, India; Department of Community Medicine, Sri Manakula Vinayagar Medical College and Hospital, Puducherry, India; Karuna Trust, Bangalore, India; Public Health Foundation of India, Gurgaon, India; The INCLEN Trust International, New Delhi, India; Indian Council of Medical Research (ICMR), Department of Health Research, Ministry of Health and Family Welfare, New Delhi, India; Department of Community Medicine, Sri Devraj Urs Medical College, Kolar, India; and Department of Community Medicine, Yenepoya Medical College, Mangalore, India. Further, the support of all the Xpert and reference laboratory technicians and microbiologists, District TB Officers and the State TB Officer is highly valued.

## Author Contributions

**Conceptualization:** Uma Shankar S., Ajay M. V. Kumar, Nikhil Srinivasapura Venkateshmurthy, Divya Nair, Reena Kingsbury, Somashekar N.

**Data curation:** Uma Shankar S., Ajay M. V. Kumar, Divya Nair, Reena Kingsbury, Padmesha R., Magesh Velu, Suganthi P., Joydev Gupta, Jameel Ahmed, Puttaswamy G., Somashekarayya Hiremath, Ravi K. Jaiswal, Rony Jose Kokkad.

**Formal analysis:** Uma Shankar S., Ajay M. V. Kumar, Divya Nair.

**Funding acquisition:** Uma Shankar S.

**Investigation:** Uma Shankar S.

**Methodology:** Uma Shankar S., Ajay M. V. Kumar, Nikhil Srinivasapura Venkateshmurthy, Divya Nair.

**Project administration:** Uma Shankar S., Somashekar N.

**Resources:** Uma Shankar S., Ajay M. V. Kumar, Somashekar N.

**Software:** Ajay M. V. Kumar, Ravi K. Jaiswal, Rony Jose Kokkad.

**Supervision:** Uma Shankar S., Nikhil Srinivasapura Venkateshmurthy, Rony Jose Kokkad, Somashekar N.

**Validation:** Divya Nair, Ravi K. Jaiswal, Rony Jose Kokkad.

**Visualization:** Uma Shankar S., Ajay M. V. Kumar, Nikhil Srinivasapura Venkateshmurthy, Reena Kingsbury.

**Writing – original draft:** Uma Shankar S., Ajay M. V. Kumar, Nikhil Srinivasapura Venkateshmurthy.

**Writing – review & editing:** Uma Shankar S., Ajay M. V. Kumar, Nikhil Srinivasapura Venkateshmurthy, Divya Nair, Reena Kingsbury, Padmesha R., Magesh Velu, Suganthi P., Joydev Gupta, Jameel Ahmed, Puttaswamy G., Somashekarayya Hiremath, Ravi K. Jaiswal, Rony Jose Kokkad, Somashekar N.

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
