## [Decision Letter · Decision Letter 0]

29 Jan 2020

PONE-D-19-25977

Implementation of the new integrated algorithm for diagnosis of drug resistant tuberculosis in Karnataka State, India: How well are we doing?

PLOS ONE

Dear Dr. Shankar S,

Thank you for submitting your manuscript to PLOS ONE. After careful consideration, we feel that it has merit but does not fully meet PLOS ONE’s publication criteria as it currently stands. Therefore, we invite you to submit a revised version of the manuscript that addresses the points raised during the review process.

In particular, concerns regarding statisical analysis should be addressed and some parts of manuscript should be clarified e.g. Introduction, as recommended by the reviewers.

We would appreciate receiving your revised manuscript by Mar 14 2020 11:59PM. To enhance the reproducibility of your results, we recommend that if applicable you deposit your laboratory protocols in protocols.io, where a protocol can be assigned its own identifier (DOI) such that it can be cited independently in the future. For instructions see: http://journals.plos.org/plosone/s/submission-guidelines#loc-laboratory-protocols

We look forward to receiving your revised manuscript.

Kind regards,

Igor Mokrousov, Ph.D., D.Sc.

Academic Editor

PLOS ONE

Journal Requirements:

3. Please amend the manuscript submission data (via Edit Submission) to include authors Reena Kingsbury, Ramamurthy Padmesha, Magesh Velu, Suganthi P, Joydev Gupta, Jameel Ahmed, Puttaswamy G, Somashekarayya Hiremath, Ravi K Jaiswal and Rony Jose Kokkad.

Reviewers' comments:

Reviewer's Responses to Questions

**Comments to the Author**

1. Is the manuscript technically sound, and do the data support the conclusions?

Reviewer #1: Partly

Reviewer #2: Partly

Reviewer #3: Yes

Reviewer #4: Yes

2. Has the statistical analysis been performed appropriately and rigorously? 

Reviewer #1: No

Reviewer #2: Yes

Reviewer #3: Yes

Reviewer #4: Yes

3. Have the authors made all data underlying the findings in their manuscript fully available?

Reviewer #1: No

Reviewer #2: Yes

Reviewer #3: Yes

Reviewer #4: Yes

4. Is the manuscript presented in an intelligible fashion and written in standard English?

Reviewer #1: Yes

Reviewer #2: Yes

Reviewer #3: Yes

Reviewer #4: Yes

5. Review Comments to the Author

Reviewer #1: Summary:

This manuscript examines a new diagnostic algorithm in Karnataka state for the identification of drug resistant TB. The authors have presented results for 1660 patients diagnosed over a two-month period in 2018, and indicate that the while there are some specimens that did not reach the second testing lab or complete the testing algorithm, overall they are encouraged by the proportion that made it through the complex algorithm. Factors associated with specimens not making it through the testing process were examined, and the turn-around-times were reported.

I enjoyed reading this manuscript and found it to be generally clear despite the challenges of describing complicated algorithms. There are however some concerns with the statistical methods used to examine factors that may be associated specimens reaching the reference labs and not completing the testing algorithm. There are some minor issues related to clarity of message and grammar/wording which I will offer suggestions as PLoS ONE does not copyedit accepted manuscripts.

Major issues:

Line 290. Unless I am misunderstanding the data, a poisson regression model would be the incorrect choice given the binary outcome variable (Not reaching vs reaching; completion vs non-completion). The measure of the association would be an odds ratio. Please revise accordingly.

Line 291. An exploratory data analysis is often necessary to conduct initially to understand the dataset and variables; however, for reporting results all variables included in a model should make sense. They should either be chosen a priori based on literature and experience, and/or because an association was observed on bivariate analysis. Could you please elaborate on what support there would be for an association between not reaching or not completing, for variables related to the patient such as age, sex and key population? Do the laboratory personnel processing specimens have access to this information and would make a choice not to send or process specimens and samples based on this information? It seems more likely that an association is possible with respect to variables related to the specimens themselves or facilities. If there is no statistical, literature, or common knowledge that would indicate that patient-level variables be considered in the multivariable model please update the analysis without these and revise accordingly. If there is clear rationale to support their inclusion please update the methods to include this information.

Table 4. Analysis for non-completion. Kolar district is no longer significant when covariates are added to the model. Did you investigate why this might be? What is the relevant confounder to the association? Given my concerns regarding the covariates included this warrants further investigation.

It does not appear there was an assessment of model fit, and there may be issues related to overfitting. There are quite a few variables included, and a some of these have many levels. Cell sizes were quite low in some instances in the non-completion model. It is recommended that the model building approach and assessment of fit be carefully considered.

Minor issues:

Abstract:

1. Line 65. Capitalize Line Probe Assay for consistency with the manuscript text.

2. Line 67. Insert the word ‘phenotypic’ before drug susceptibility testing

3. Line 75. Please change ‘till’ to ‘until’ (more formal for scientific writing)

4. Line 78. Were there any duplicate patients in the 1660 (more than one specimen/patient) if so then patients should be updated to ‘specimens’ here, and throughout the manuscript as necessary. It would also be helpful to indicate in the methods or results if each patient is represented by a single specimen, or if not please report the number of patients with multiple specimens and an indication of how many specimens/patient were represented in this study.

5. While I agree that putting this complex algorithm into place is encouraging and having a considerable proportion of specimens make it through testing, I am also concerned that only about a quarter of RIF resistant TB completed the testing algorithm – which is the TB that is most in need of full testing. It would be beneficial to include a few words in the abstract and again in the manuscript indicating that while encouraging there is also significant concern about this.

Introduction:

1. Line 113. Please add the word ‘phenotypic’ in front of drug susceptibility

2. Line 118. A better word for ‘heartening’ in scientific writing may be ‘encouraging’

3. Line 126. Can you clarify here if by non-diagnosis it is meant that resistance was not tested/diagnosed? If patients weren’t diagnosed at all they wouldn’t be treated and therefore not part of the statistic for treatment failure.

4. Line 184. Reference #15 does not link to the pdf for the correct province, please update.

5. Line 185. The reference #15 does not appear to contain the statistic for prevalence, please update the reference to the correct one and ensure that the statistic reported is indeed prevalence rather than incidence.

Methods:

1. Line 195-196. The company that produces the products listed should be included, Hain Lifescience, BD Biosciences BACTEC, and which version of the line probe assays were used.

2. Line 199. Could you elaborate on the human carriers and how this differs from a courier service?

3. Line 215. It may be better to specify ‘one day’ rather than ‘a day’.

4. Line 219. Perhaps reword to read: ‘testing is required, and the second sample…’

5. Line 224. Please specify in the methods or introduction which second line drugs are being tested, not all readers may be familiar with the Hain assay and the version number was not indicated.

6. Line 224. I assume that further testing is for those that were resistant to second line drugs in the SL-LPA? Perhaps clarify this in the sentence.

7. Line 229. It appears that a word is missing. Either ‘a sample’ or ‘their sample’

8. Line 230. Could you clarify the use of poly resistant treatment? As I understand the sentence these cases are RIF-S and INH-R, so initiation of poly-resistant treatment would be indicated by FL-LPA indicating resistance to EMB or PZA.

9. Line 245. I assume the data was collected at the time of specimen collection and testing, and here you mean the data was extracted between February and April 2019 from the laboratory registers?

10. Line 274-275. Quotation marks are not needed here.

11. How were the laboratories selected for this study? Was there specific criteria?

Results:

1. Line 309. This is the first time “key population group” has been introduced and does not include an explanation of what this represents. Information regarding this should be included in the methods as it is not a standard variable.

2. Line 310. In the context of this sentence, it should be ‘persons living with HIV’

3. It is not clear what happened to the specimens with indeterminant or no information? – Where do they fit in the algorithm? What was the outcome? Indicate in the methods sections how these were handled for the analysis.

4. Line 347. It is stated that culture was done for 41 samples. however, in Figure 2 it indicates culture was done for 43 samples?

5. Lines 361-363. This sentence is a bit hard to follow. Perhaps something along the lines of

“on eight samples, under the following SL-LPA conditions: (i) resistance, (ii) was not done, or (iii) no result.”

6. Line 373. Based on Figs 2 and 3, the number that reached the laboratory was 1170+35 = 1205 and therefore n=455 did not reach the laboratory? which also affects the calculation of those that completed the algorithm. Please clarify.

7. Line 379. Please report the precise number of samples that reached within one day for the results section.

8. Lines 380 and 383. IQR should be included with the median.

Discussion:

1. Line 451. It may be helpful to provide a reference here regarding the standard or recommended turn around times to support this statement that there were no major delays.

2. Line 452. Please expand briefly on the SL-LPA testing delays.

3. Line 455. How far away?

4. Line 457. One-fourth is not commonly used in this context. ‘Approximately 25%’ or ‘approximately one quarter’

5. Lines 457-463. What are the reasons that extrapulmonary specimens do not get tested? Do you have any recommendations on how to improve this?

6. Line 465. Intrigued is an interesting choice of word here. I would have thought surprised or dismayed.

7. Lines 465-470. Is this similar or different to the findings of other studies? A common issue that RIF-R samples are not submitted for further testing? This result is a major finding and should be discussed further and stressed as important. Instead of ‘This needs further investigation’ at the very least something like ‘This is an important issue and requires further investigation’.

8. Lines 472. How was is better?

9. Line 473. Please reword to: “This may be due to the presence of a system of…”

10. Line 475. Instead of gaps, “proportion of specimens that did not reach”

11. Line 479. You may want to include the percentage here to highlight how excellent they were.

12. Line 480-483. Could you please clarify why delays related to culture would result in non-completion of the algorithm?

13. Line 480. If evidence of this was not provided as a result for why there were delays than this is speculation? If you do not have results demonstrating this please rephrase to “This may be explained”

14. Line 481. Correct wording to ‘…must be cultured before’ or ‘…require culture before’

15. Line 489. See previous comment. If evidence is not presented in the results, “This may explain” as it is an assumption.

16. Lines 501-503. Following STROBE is not a strength of the study, but of the manuscript writing. Please remove.

17. Line 511. Briefly expand on this, and how it limits your ability to recommend changes/improvements. What information would be needed to inform these gaps and how would you propose to gather it?

18. Lines 515-519. Here you have given specific examples of things to improve; however, there were no results proving that these were the specific issues to be addressed. Please revise.

19. Line 529. ‘Approximately’ is a preferred word to ‘about’. This sentence could be more concise.

20. Line 531. ‘gaps’ should be expanded here – delays? not reaching the reference laboratory? not completing the algorithm?

21. Lines 532-534. This should be included in the acknowledgements section rather than conclusions.

Tables

Please be consistent across tables with capitalization and variable names. e.g. Table 2 non-key population; Table 4 Not Key population. Similarly, use consistent term for unavailable data for each variable. ‘Missing’ or ‘Not recorded’ or ‘Not available’. Also, with the site and laboratory.

Table 2.

1. Under key population: reword to ‘persons living with HIV’

2. Under key population: Footnote should indicate what population is represented in ‘Others’

3. For Specimen condition of receipt at Xpert lab – these categories only apply to sputum specimens? The table should reflect this.

Table 3.

1. The last line FL-LPA result was resistant – to any first line drug?

Table 4.

1. Male reference is not indicated in aRR column

2. There is no footnote for the abbreviation PLHIV

3. The N and (%) columns presumably refer the those that did not reach the laboratory. Please clarify this in the column header.

Table 5.

1. As in Table 4, please be specific for the column header N (%)

2. Age 0-14 is not an appropriate reference given that the N = 0 for non-completion.

3. Please indicate what the NA values represent.

4. Under Specimen type, remove the word sample for Extra-pulmonary

Figures

Figure 2.

1. The denominator for culture done does not appear to add up. 39+21+43+4 = 107

2. For the 3 Xpert RIF-S samples that were then RIF-R as the reference laboratory what was the pattern of INH resistant? This could be included as a footnote.

Reviewer #2: Overall comments: Thank you for the opportunity to review this manuscript. It is a well written manuscript that addresses a topic that is of fundamental importance to TB care in India.

Abstract: No major comments

Introduction:

• Line 106: Is this prevalence of MDR-TB among all cases, if so I think specify.

• Line 108: Spell out XDR TB at first use

• Line 112: I think it would be good to have a brief explanation of what PMDT is.

• Line 116-117: is this statistic of 29% from India? Pls kindly clarify so that the context is clear.

• Line 126: You talk about non diagnosis as being one of the reasons for low treatment success but treatment success is really only measured for diagnosed cases, pls clarify.

• Paragraph starting at line 130: I would have liked to know about more about the rollout of Xpert in India, can you provide a couple more sentences about this including the dates of rollout and how quickly it happened?

• Line 133: When did the policy of universal DST start? A date would be helpful.

Methods:

• Line 183: I think this sentence about the population size needs a reference.

• You mention the human carriers or couriers in lines 199-200 and then again in lines 208-209 which I think is repetitious.

• Line 216-217: Is this second sample also tested using Xpert or LPA?

• Line 223: I think it would be better to say “If additional resistance” rather than “If resistant” as I think this is what is meant, i.e. if there is additional resistance then a pre XDR or XDR regimen is started.

• I was wondering why a mobile phone number was used as the second method of identifying people rather than the name-age-sex combination which may be more unique. How well does a mobile phone number identify the user? Has this method been previously validated for matching people in population based studies? I think this needs further discussion and justification.

• In Table 1 I think some additional clarity is needed, i.e. for the third to fifth bullet points what is the resistance or sensitivity to? I think some additional detail is needed here. It should also be clear why completion of the diagnostic algorithm was constructed the way it was including having the denominator start at the reference laboratory as the diagnostic algorithm actually seems to start before then, i.e. in the Xpert laboratory.

Results:

Overall the results section was well constructed and clear. My main comment relates to the numbers and Figures 2 and 3 and the definition of having completed the diagnostic algorithm. For Figure 2 I am not 100% sure how you got the figure of 103 in the culture done box, should this be 107 (i.e. 4 plus 43 plus 21 plus 39)? If I follow the lines on all of the boxes that lead the culture done box I get 107 instead of 103. And I wondered why the people who are susceptible or who had culture are the only ones who are deemed eligible to have completed the algorithm? If there is resistance on FL LPA and then that person goes on to have the appropriate tests, they have also completed the algorithm haven’t they? In Figure 3 should there be a lone from the box results not available to the box culture done so that the total is 26 and not 24? I also wondered if your denominators should really be 1590 and 64 rather than the denominators that you have as this is where the algorithm starts. For Figure 3 I also wondered if the people who completed the algorithm should be the 14 who had SL LPA and then any additional people who had culture when it was indicated. I think the 9 people who were FQ and SLI susceptible are include in the numerator of 17 but if you are resistant doesn’t it also mean that you have completed the algorithm?

Discussion:

• Line 461: I think you should reference the “previous studies” referred to here and as a general comment I think there could be more use of other studies in the Discussion section as it mainly focuses on the findings of the study rather than comparing and contrasting with other literature from India, the region or elsewhere. There is one study mentioned in lines 461-463 but it is not clear what date this was and it is a study on EPTB so may not be directly comparable to your overall sample as the majority of your sample were PTB (although admittedly it does seem that EPTB samples were less likely to be referred to the reference laboratory).

• I think it could be emphasized a bit more the loss of specimens going from the Xpert lab to the reference lab and the implications of this. I think you could also emphasise the losses for the RR cases as well as these are the very cases that you would want to know have completed the diagnostic algorithm.

• Under the section on Strengths you talk about sensitivity and specificity of individual tests but you did not do this so I would recommend leaving this out

Reviewer #3: Comments:

The subject of the manuscript has merit and describes important findings related to Drugs resistant TB diagnostic algorithm under routine programme settings in India. The authors may address following queries to strengthen the manuscript.

Major concerns:

1) Introduction- Introduction may need restructuring.

- The first paragraph seems general and it discusses about prevention, diagnosis and treatment while the manuscript is only about diagnostic cascade.

- Line 124-128: It describes about treatment success and its linkage to delay in diagnosis. This section could be mentioned in the discussion.

2) Methods:

- Line 241: How were these ten districts selected for the study? Please provide some information.

- The study mentions about gap in UDST, however the last steps of the algorithm is considered as Culture done. I wonder if it must end at DST level (for how many had DST was done). Though the operational definition mentions the algorithm finishes at culture done, the authors may want to describe about this concern in the manuscript.

3) Results:

- Line 347-348: Please check if the total eligible for culture was 107 instead of 103.

4) Discussion:

- Line 448-449: Since the integrated DR-TB diagnostic algorithm is specific for India, you may tone down the first statement.

- Line 457-459 seems a repeat of the results, please review.

- Line 489: when the authors mention about the delays, they may consider that availability of dates for SL-LPA was quite low. The claims could be toned down.

Minor concerns:

- Please update the references. For example, Line 105 must include the recent literature (Global TB Report 2019).

- Line 167: no systematic assessment.. do we mean .. in India? If yes, you may want to mention this.

- Line 199: change ‘…were transported..’ to ‘..are transported..’

- Line 200-202: This should be mentioned under the Programme implementation part of the Methods section.

- Line 204-237: The integrated diagnostic algorithm: Can the authors summarize the section, as the same is described in the Figure 1.

- Line 251- double entry and validation, wherever possible: please explain where it was carried out and where it was not possible.

- Line 309: Were the key population mutually exclusive group. If someone was urban slum dweller and PLHIV, in which category they were considered?

- Table 2: It is good to mention in the title of the table that 13 Xpert laboratories were included in the study

- Line 461: Please add reference to the statement.

- Line 474: In Methods it was mentioned that the NGO was working in all selected districts of study, please review and change the statement.

END

Reviewer #4: I wish to congratulate the authors on this very clear and helpful paper. It is a transparent analysis of an operational challenge in TB control, which will be of benefit to others working in the same field. I however do have a small number of concerns that I would suggest the authors address, before recommending this manuscript for publication:

1) This study on the performance of the UDST in Karnataka was conducted only a few months after its implementation (data from July-August, for a system implemented in April). Could the authors comment on whether the results are likely to be affected by the study being conducted in this early phase? Could there be "teething troubles" with the UDST, or conversely could there be an ambitious start, which then deteriorates over time?

2) In the same vein: the study was conducted only over 2 months (July-August) - do the authors anticipate any seasonality in the performance of the UDST?

3) The study hinges to a large extent on the matching algorithm that was used between database 1 and database 2, relying on the Nikshay number, phone number, and a name/age/sex match. This is a commendable effort, but it is not without risk. I would recommend that the authors report on how well the matching worked (which proportion was matched on Nikshay, which on phone number, etc.); possibly as supplemental material. Additionally, if I understand well, all entries in database 2 should have a corresponding match in database 1 (as no patients would end up directly at the reference lab) - any "unmatched" individuals in database 2 would therefore represent a measure of how many incorrect matchings resulted from the algorithm, and this may be worth reporting on.

4) Line 290-291: I am not entirely clear on the "exploratory approach" applied, and/or why *all* factors were included in the adjusted analysis.

5) While I appreciate the various ethics reviews done, I would suggest to expand on how patient confidentiality was protected, given the extensive use of phone numbers and names in this study.

6) Line 348-350 ("Thus, a total of 1106 (95%) of rifampicin-sensitive TB patients were considered to have completed the diagnostic algorithm.") is perhaps phrased a bit too optimistically, given that a large proportion of RS TB samples did not even show up at the reference lab. I would suggest correcting to "(...) of rifampicin-sensitive TB patients whose samples were successfully received at the reference laboratory were considered to have completed the diagnostic algorithm." Same comment for line 363-364.

7) On a minor note: one of the percentages in table 3 is incorrect (9.1 should read 91.4).

8) In the methods section the authors refer to "selected districts", and in the limitations to "selected laboratories". Could they clarify the selection process, and which criteria were used?

6. PLOS authors have the option to publish the peer review history of their article (what does this mean?). If published, this will include your full peer review and any attached files.

Reviewer #1: No

Reviewer #2: No

Reviewer #3: No

Reviewer #4: No

---

## [Author Response · Author response to Decision Letter 0]

31 Aug 2020

28th July, 2020

Bangalore, India

To 

The Editor,

PLoS ONE

Dear Editor,

We thank you and the reviewers for taking time to review our paper and provide constructive comments and suggestions. We have been through the comments and provide here a point-by-point response. We have revised the paper accordingly and submit two versions – one with track changes reflecting all the changes and another, a clean version. We have also used the opportunity to read the paper in its entirety for editorial corrections, as and where required. As a result of this process, the manuscript has improved considerably in clarity and readability.

I hope this version meets your satisfaction. If there are any further comments, we will be happy to address them.

Best wishes,

Dr Uma Shankar S, on behalf of the co-authors 

Reviewer #1: Summary:

This manuscript examines a new diagnostic algorithm in Karnataka state for the identification of drug resistant TB. The authors have presented results for 1660 patients diagnosed over a two-month period in 2018, and indicate that the while there are some specimens that did not reach the second testing lab or complete the testing algorithm, overall they are encouraged by the proportion that made it through the complex algorithm. Factors associated with specimens not making it through the testing process were examined, and the turn-around-times were reported.

I enjoyed reading this manuscript and found it to be generally clear despite the challenges of describing complicated algorithms. There are however some concerns with the statistical methods used to examine factors that may be associated specimens reaching the reference labs and not completing the testing algorithm. There are some minor issues related to clarity of message and grammar/wording which I will offer suggestions as PLoS ONE does not copyedit accepted manuscripts.

Response to the reviewer: Thank you for the appreciation. We have addressed all the comments below, point-by-point.

Major issues:

1. Line 290. Unless I am misunderstanding the data, a poisson regression model would be the incorrect choice given the binary outcome variable (Not reaching vs reaching; completion vs non-completion). The measure of the association would be an odds ratio. Please revise accordingly.

Response to the reviewer: We thank the reviewer for the comment. This is a cohort study and it enables direct calculation of risk ratio, the most preferred effect measure in epidemiology. While odds ratios can also be calculated, they often overestimate associations, especially when the outcomes are common (as is the case in this study with 27% non-reaching). We also think relative risk (compared to odds ratio) would be easy to communicate to policy makers and program managers and also for them to understand. (1,2). There are multiple ways to arrive at adjusted relative risk and we have opted for the Poisson model as no convergence was obtained in the log binomial model. We have provided references in support of our stance here. 

Changes to the manuscript: No changes made to the manuscript

References: 

1. UCLA Institute for Digital Research and Education. How can I estimate relative risk using glm for common outcomes in cohort studies? | stata FAQ [Internet]. Institute for Digital Research and Education. [cited 2020 June 18]. Available from: https://stats.idre.ucla.edu/stata/ faq/how-can-i-estimate-relative-risk-using-glm-forcommon-outcomes-in-cohort-studies/.

2. McNutt L-A, Wu C, Xue X, et al. Estimating the relative risk in cohort studies and clinical trials of common outcomes. Am J Epidemiol [Internet].2003;157:940–943. [cited 2020 June 18].

2. Line 291. An exploratory data analysis is often necessary to conduct initially to understand the dataset and variables; however, for reporting results all variables included in a model should make sense. They should either be chosen a priori based on literature and experience, and/or because an association was observed on bivariate analysis. Could you please elaborate on what support there would be for an association between not reaching or not completing, for variables related to the patient such as age, sex and key population? Do the laboratory personnel processing specimens have access to this information and would make a choice not to send or process specimens and samples based on this information? It seems more likely that an association is possible with respect to variables related to the specimens themselves or facilities. If there is no statistical, literature, or common knowledge that would indicate that patient-level variables be considered in the multivariable model please update the analysis without these and revise accordingly. If there is clear rationale to support their inclusion, please update the methods to include this information.

Response to the reviewer: 

We thank the reviewer for the comment. Regarding association of age with completion of diagnostic algorithm, there could be a significant association due to the variations in quality and quantity of the sample being collected. For e.g., it is difficult to collect mucopurulent sample from children, which needs to be obtained after deep breathing and intense coughing and the quantity can also be less when collected from children. Similarly, key population (tobacco users, PLHIV etc.), are relevant because the policies for TB management identifies them so. We would like to submit that these are standard variables which could act as confounders in the analysis. Confounders may not be significant in unadjusted analysis, but they may be unmasked during multivariable analysis. We agree that associations may or may not be present but exploring these variables help in identifying population groups which may require intervention. 

Changes made in the manuscript: No changes were done in the manuscript

3. Table 4. Analysis for non-completion. Kolar district is no longer significant when covariates are added to the model. Did you investigate why this might be? What is the relevant confounder to the association? Given my concerns regarding the covariates included this warrants further investigation.

Response to the reviewer: We thank the reviewer for the comment. We wish to clarify that Kolar district was significantly associated with samples ‘not reaching’ the reference laboratory and not with non-completion of the diagnostic algorithm. There might be many reasons for this including differences in distributions of other key variables such as the proportion of rifampicin resistant samples, EP TB samples or absence of efficient sample transportation mechanisms. Since we did not study the reasons, we are not able to postulate why Kolar was not significant in adjusted analysis. 

Changes made in the manuscript: No changes were done in the manuscript

4. It does not appear there was an assessment of model fit, and there may be issues related to overfitting. There are quite a few variables included, and a some of these have many levels. Cell sizes were quite low in some instances in the non-completion model. It is recommended that the model building approach and assessment of fit be carefully considered.

Response to the reviewer: We thank the reviewer for the comment. The approach to analysis was exploratory and we wanted to identify as many factors as possible and also identify such factors which would help guide the programme managers to take informed decisions wherever required. Since there are no studies from India and really very few from elsewhere in the world which explored the completion of a complex diagnostic algorithm having a cascade of tests to be done at two different laboratories, a statistical approach of exploring the goodness of fit was not done initially. We however ran the goodness of fit test (estat gof command in Stata) for our model and found that the model fits well (p value 0.957) 

Changes in the manuscript: No changes were done in the manuscript

Minor issues:

Abstract:

1. Line 65. Capitalize Line Probe Assay for consistency with the manuscript text.

Response to the reviewer: We thank the reviewer for the suggestion. We have made the correction. 

Changes in the manuscript: Please refer to lines 64, 84 and 119 of ‘Revised Manuscript with Track Changes’

2. Line 67. Insert the word ‘phenotypic’ before drug susceptibility testing

Response to the reviewer: We thank the reviewer for the suggestion. We have made the correction.

Changes in the manuscript: Please refer to the line no. 66 of ‘Revised Manuscript with Track Changes’

3. Line 75. Please change ‘till’ to ‘until’ (more formal for scientific writing)

Response to the reviewer: We thank the reviewer for the suggestion. We have made the correction. 

Changes in the manuscript: Correction reflected in Line 74 of ‘Revised Manuscript with Track Changes’

4. Line 78. Were there any duplicate patients in the 1660 (more than one specimen/patient) if so then patients should be updated to ‘specimens’ here, and throughout the manuscript as necessary. It would also be helpful to indicate in the methods or results if each patient is represented by a single specimen, or if not please report the number of patients with multiple specimens and an indication of how many specimens/patient were represented in this study.

Response to the reviewer: We thank the reviewer for the suggestion. The number 1660 represents the number of patients. Rarely, multiple specimens were collected from the same patient, say whenever a new TB patient was diagnosed with rifampicin resistance (refer to specific setting in the paper) or whenever a sample container was damaged/leaking on receipt at the laboratories, a second sample was requested. However, we considered only the latest results for a given patient, even if there were more than one sample tested. Hence, there were 1660 unique patients.

Changes to the manuscript: None.

5. While I agree that putting this complex algorithm into place is encouraging and having a considerable proportion of specimens make it through testing, I am also concerned that only about a quarter of RIF resistant TB completed the testing algorithm – which is the TB that is most in need of full testing. It would be beneficial to include a few words in the abstract and again in the manuscript indicating that while encouraging there is also significant concern about this.

Response to the reviewer: We thank the reviewer for the suggestion. We have made the correction. 

Changes in the manuscript: Please see the changes made in Lines 87-88 and 551-553 of ‘Revised Manuscript with Track Changes’

Introduction:

1. Line 113. Please add the word ‘phenotypic’ in front of drug susceptibility

Response to the reviewer: We thank the reviewer for the suggestion. We have made the correction. 

Correction to the manuscript. Please see the changes made in line no. 66 and 116

Line 118. A better word for ‘heartening’ in scientific writing may be ‘encouraging’

Response to the reviewer: We thank the reviewer for the suggestion. We have made the correction. 

Changes in the manuscript: Edits reflected in Line 124 of ‘Revised Manuscript with Track Changes’

2. Line 126. Can you clarify here if by non-diagnosis it is meant that resistance was not tested/diagnosed? If patients weren’t diagnosed at all they wouldn’t be treated and therefore not part of the statistic for treatment failure.

Response to the reviewer: We thank the reviewer for the suggestion. “Non-diagnosis” has been removed. What we meant was delays in diagnosis and treatment leading to poor outcomes.

Changes in the manuscript: Edits reflected in line 134 of ‘Revised Manuscript with Track Changes’.

4. Reference #15 does not link to the pdf for the correct province, please update.

Response to the reviewer: We thank the reviewer for the suggestion. We have made the correction. 

Changes in the manuscript: Reference Number 15 updated.

5. Line 185. The reference #15 does not appear to contain the statistic for prevalence, please update the reference to the correct one and ensure that the statistic reported is indeed prevalence rather than incidence.

Response to the reviewer: We thank the reviewer for the suggestion. The prevalence of TB according to NFHS-4 is 180/100,000 population in Karnataka. This figure is reported in Table 77, Page 130 of the referenced document.

Changes in the manuscript: Reference Number 15 updated.

Methods:

1. Line 195-196. The company that produces the products listed should be included, Hain Lifescience, BD Biosciences BACTEC, and which version of the line probe assays were used.

Response to the reviewer: We thank the reviewer for the suggestion. 

Changes in the manuscript: Edits reflected in Line 246-250 of ‘Revised Manuscript with Track Changes’

2. Line 199. Could you elaborate on the human carriers and how this differs from a courier service?

Response to the reviewer: We thank the reviewer for the comment and the suggestion. Human carriers are the employees of the health facility or programme staff stationed at the facility. They may be multipurpose health workers, TB-Health Visitors, etc. who might be entrusted with specimen transport on days that they are able to visit the reference laboratory during their routine work-related trips. As a routine, courier services specialising in transport of samples are used. This has been explained in the manuscript.

Changes in the manuscript: Edits reflected in Line 250-253 of ‘Revised Manuscript with Track Changes’.

3. Line 215. It may be better to specify ‘one day’ rather than ‘a day’.

Response to the reviewer: We thank the reviewer for the suggestion. We have made the change in the manuscript. 

Changes in the manuscript: Edits reflected in Line 268 of ‘Revised Manuscript with Track Changes’.

4. Line 219. Perhaps reword to read: ‘testing is required, and the second sample…’

Response to the reviewer: We thank the reviewer for the suggestion. We have made the change in the manuscript. 

Changes in the manuscript: Edits reflected in Line 272 of ‘Revised Manuscript with Track Changes’

5. Line 224. Please specify in the methods or introduction which second line drugs are being tested, not all readers may be familiar with the Hain assay and the version number was not indicated.

Response to the reviewer: We thank the reviewer for the suggestion. We have added he names of drugs. SL-LPA tests for resistance to fluoroquinolone or second line class of drugs. Liquid CDST tests for resistance to levofloxacin, moxifloxacin, linezolid, kanamycin, amikacin. Version number has now been indicated along with the brand name in the previous section. 

Changes in the manuscript: Edits reflected in Lines 281-282 of ‘Revised Manuscript with Track Changes’

6. Line 224. I assume that further testing is for those that were resistant to second line drugs in the SL-LPA? Perhaps clarify this in the sentence.

Response to the reviewer: We thank the reviewer for the suggestion. Yes this assumption is correct and changes have been made to make this clear.

Changes in the manuscript: Edit reflected in Line 288 of ‘Revised Manuscript with Track Changes’

7. Line 229. It appears that a word is missing. Either ‘a sample’ or ‘their sample’

Response to the reviewer: Thanks for the reviewing and suggesting. Correction done.

Changes in the manuscript: Edit reflected in Line 286 of ‘Revised Manuscript with Track Changes’

8. Line 230. Could you clarify the use of poly resistant treatment? As I understand the sentence these cases are RIF-S and INH-R, so initiation of poly-resistant treatment would be indicated by FL-LPA indicating resistance to EMB or PZA.

Response to the reviewer: We thank the reviewer for the suggestion. A Poly drug resistant TB patient is one whose biological specimen is resistant to more than one first-line anti-TB drug, other than both INH and RIF. In operational terms based on the FL drugs for which susceptibility testing is carried out, this translates to patients who are RIF susceptible, INH resistant and whose susceptibility to streptomycin, ethambutol and pyrazinamide is not known. These patients are treated using the same regimen as INH mono resistant TB as per the RNTCP guidelines.

A phrase is added- “…for isoniazid resistant TB (Wherein resistance to the other first-line anti-TB drugs is not done/not known) to clarify the term “poly resistant TB.

Changes in the manuscript: Edit reflected in Lines 288 of ‘Revised Manuscript with Track Changes’

9. Line 245. I assume the data was collected at the time of specimen collection and testing, and here you mean the data was extracted between February and April 2019 from the laboratory registers?

Response to the reviewer: We thank the reviewer for the suggestion. Yes, the data was extracted between February and April 2019 from the laboratory registers. Change made in the manuscript to this effect.

Changes in the manuscript: Edit reflected in Line 304-305 of ‘Revised Manuscript with Track Changes’

10. Line 274-275. Quotation marks are not needed here.

Response to the reviewer: We thank the reviewer for the suggestion. We have made the change in the manuscript. 

Changes in the manuscript: Edits reflected in Lines 340-343 of ‘Revised Manuscript with Track Changes’

11. How were the laboratories selected for this study? Was there specific criteria?

Response to the reviewer: The districts were selected conveniently based on feasibility of data collection. Within a district all the Xpert/CBNAAT sites were included.

Changes to the manuscript: Edits reflected in Lines 299-301 of ‘Revised Manuscript with Track Changes’

Results:

1. Line 309. This is the first time “key population group” has been introduced and does not include an explanation of what this represents. Information regarding this should be included in the methods as it is not a standard variable.

Response to the reviewer: We thank the reviewer for the suggestion. Data captured in the laboratory registers kept at the Xpert and reference laboratories, including key population is described briefly at the end of the section on ‘The integrated diagnostic algorithm’. Key population is the term used in RNTCP for specific population groups which are known to be at high risk for TB- tobacco users, prisoners, migrants, PLHIV etc. This field is included in all RNTCP documents like patients’ treatment card and different laboratory registers.

Changes in the manuscript: Edits reflected in Lines 310-312 of ‘Revised Manuscript with Track Changes’

2. Line 310. In the context of this sentence, it should be ‘persons living with HIV’

Response to the reviewer: We thank the reviewer for the suggestion. The national programme documents in India (http://naco.gov.in/documents/annual-reports Page no 335 of the Annual report 2015-16 of the National AIDS Control Organization of India) mention ‘people living with HIV’ and not “persons living with HIV”. Hence, we would like to use ‘people living with HIV’ because this is also the term used in many international documents including those by the WHO (https://www.who.int/news-room/fact-sheets/detail/hiv-aids).

Changes in the manuscript: None

3. It is not clear what happened to the specimens with indeterminant or no information? – Where do they fit in the algorithm? What was the outcome? Indicate in the methods sections how these were handled for the analysis.

Response to the reviewer: We thank the reviewer for the suggestion. We had designed the data capturing tool to collect information on the final test results in case tests were performed more than once using a diagnostic technology at the same laboratory. So, indeterminate and no results mean that this was the final result for a given test whether it was tested once or more, and no further information was available. The outcome has been defined in the section, Patients completing the diagnostic algorithm, of the Operational definitions table.

Changes in the manuscript: Edits reflected in Lines 274-275 of ‘Revised Manuscript with Track Changes’. 

4. Line 347. It is stated that culture was done for 41 samples. however, in Figure 2 it indicates culture was done for 43 samples?

Response to the reviewer: We thank the reviewer for the suggestion. The actual figure is 41 out of 107 and the same has been updated in the fig 2.

Changes in the manuscript: Edits reflected in Lines 418 of ‘Revised Manuscript with Track Changes’

5. Lines 361-363. This sentence is a bit hard to follow. Perhaps something along the lines of “on eight samples, under the following SL-LPA conditions: (i) resistance, (ii) was not done, or (iii) no result.”

Response to the reviewer: We thank the reviewer for the suggestion. The sentence has been modified. 

Changes in the manuscript: Edits reflected in Lines 437-438 of ‘Revised Manuscript with Track Changes’

6. Line 373. Based on Figs 2 and 3, the number that reached the laboratory was 1170+35 = 1205 and therefore n=455 did not reach the laboratory? which also affects the calculation of those that completed the algorithm. Please clarify.

Response to the reviewer: We thank the reviewer for the suggestion. Please note that 6 patients results were neither sensitive nor resistant, of which 4 had indeterminate results and 2 had no results. Of the six specimens, three reached the reference laboratories. Therefore, the number non-reaching was 452 and not 455. Operational definitions specify that culture done on samples with indeterminate or no results on either FL-LPA or SL-LPA will be considered as completed the algorithm.

Changes to the manuscript: None.

7. Line 379. Please report the precise number of samples that reached within one day for the results section.

Response to the reviewer: Thanks for the reviewing and suggesting. A total of 1250/1554 (80.4%) samples reached the laboratory within one day. 

Changes to the manuscript: Edits reflected in Lines 454 of ‘Revised Manuscript with Track Changes’

8. Lines 380 and 383. IQR should be included with the median.

Response to the reviewer: We thank the reviewer for the suggestion. IQR has been added.

Changes in the manuscript: Edits reflected in Lines 457,459-461 of ‘Revised Manuscript with Track Changes’

Discussion:

1. Line 451. It may be helpful to provide a reference here regarding the standard or recommended turnaround times to support this statement that there were no major delays.

Response to the reviewer: We thank the reviewer for the suggestion. There are no reference standards for turnaround times at every step of the cascade, except for the turnaround time at the laboratory that the results should be reported within two hours for Xpert® MTB/RIF, 72 hours for LPAs and within 42 days for liquid culture and phenotypic DST by BD MGIT™. We also cannot compare with other studies, as there are no studies done in India on implementation of this integrated algorithm. The authors feel that the delays noted in this study are not “major” based on their experience of working with the TB programme in India for more than 15 years.

Changes in the manuscript: Edits reflected in Lines 294-295 of ‘Revised Manuscript with Track Changes’

2. Line 452. Please expand briefly on the SL-LPA testing delays.

Response to the reviewer: We thank the reviewer for the suggestion. SL-LPA testing delays may be related to limitations of the first-generation SL-LPA technology with high rates of indeterminate or invalid results leading to non-availability of results, requiring the specimens to be re-tested multiple times before obtaining a valid result. Since no exploration of the reasons were done, we are not in a position to offer an in-depth explanation for the delays.

Changes in the manuscript: Reasons are discussed in Lines 590-593 of ‘Revised Manuscript with Track Changes’

3. Line 455. How far away?

Response to the reviewer: We thank the reviewer for the suggestion. A line explaining the same has been added. The distance between the Xpert laboratory and the reference laboratory ranged from as close as 1 km to as far as 280 kilometres. 

Changes in the manuscript: Edits reflected in Lines 539-540 of ‘Revised Manuscript with Track Changes’

4. Line 457. One-fourth is not commonly used in this context. ‘Approximately 25%’ or ‘approximately one quarter’

Response to the reviewer: We thank the reviewer for the suggestion. We have made the change in the manuscript. 

Changes in the manuscript: Edits reflected in Line 551-552 of ‘Revised Manuscript with Track Changes’

5. Lines 457-463. What are the reasons that extrapulmonary specimens do not get tested? Do you have any recommendations on how to improve this?

Response to the reviewer: We thank the reviewer for the suggestion. Adequate sensitization of the surgeons involved in the collection of extra-pulmonary specimens on the methods and volume/size of the sample required, mechanisms for early transportation and concentration of the specimen could help increase the testing rates. Recommendations added.

Changes in the manuscript: Edits reflected in Lines 552-557 of ‘Revised Manuscript with Track Changes’

6. Line 465. Intrigued is an interesting choice of word here. I would have thought surprised or dismayed.

Response to the reviewer: We thank the reviewer for the suggestion. A change has been made in the manuscript. 

Changes in the manuscript: Edits reflected in Line 563 of ‘Revised Manuscript with Track Changes’

7. Lines 465-470. Is this similar or different to the findings of other studies? A common issue that RIF-R samples are not submitted for further testing? This result is a major finding and should be discussed further and stressed as important. Instead of ‘This needs further investigation’ at the very least something like ‘This is an important issue and requires further investigation’.

Response to the reviewer: We thank the reviewer for the suggestion. The possible reasons for this finding are discussed and a statement “This needs to be addressed on priority and requires further investigation.” has been added

Changes in the manuscript: Edits reflected in Line 592-593 of ‘Revised Manuscript with Track Changes’

8. Lines 472. How was is better?

Response to the reviewer: We thank the reviewer for the suggestion. The specimens reaching the reference laboratory from Xpert laboratories situated in districts closer to Bangalore city were better, probably due to reduced distance and sample transportation services provided by an NGO. 

Changes in the manuscript: Edits reflected in Line 571-573 of ‘Revised Manuscript with Track Changes’

9. Line 473. Please reword to: “This may be due to the presence of a system of…”

Response to the reviewer: We thank the reviewer for the suggestion. Correction may not be required as one reason for better transportation is added in the earlier line.

Changes in the manuscript: None

10. Line 475. Instead of gaps, “proportion of specimens that did not reach”

Response to the reviewer: We thank the reviewer for the suggestion. A change has been made in the manuscript.

Changes in the manuscript: Edits reflected in Line 573-574 of ‘Revised Manuscript with Track Changes’

11. Line 479. You may want to include the percentage here to highlight how excellent they were.

Response to the reviewer: We thank the reviewer for the suggestion. Percentages have been included.

Changes in the manuscript: Edits reflected in Line 578 of ‘Revised Manuscript with Track Changes’

12. Line 480-483. Could you please clarify why delays related to culture would result in non-completion of the algorithm?

Response to the reviewer: We thank the reviewer for the suggestion. Smear microscopy negative specimens must be cultured before testing by LPA. All cultures may not yield growth which may require recollection of samples, leading to non-completion of the diagnostic algorithm. We did collect data from the reference labs for a period of one month from the last date of the sample received at the Xpert labs. If culture was not done within that period, we assumed that it was not done at all. It is possible that some cultures might have been conducted beyond the one-month cut-off we had in the study. We though think (based on our programme experience) that such instances are very rare. 

Changes in the manuscript: Edits reflected in Lines 580-585 of ‘Revised Manuscript with Track Changes’

13. Line 480. If evidence of this was not provided as a result for why there were delays than this is speculation? If you do not have results demonstrating this please rephrase to “This may be explained”

Response to the reviewer: We thank the reviewer for the suggestion. 

A change has been made in the manuscript Changes in the manuscript: Edits reflected in Lines 590 of ‘Revised Manuscript with Track Changes’

14. Line 481. Correct wording to ‘…must be cultured before’ or ‘…require culture before’

Response to the reviewer: We thank the reviewer for the suggestion. A change has been made in the manuscript 

Changes in the manuscript: Edits reflected in Lines 580-581 of ‘Revised Manuscript with Track Changes’

15. Line 489. See previous comment. If evidence is not presented in the results, “This may explain” as it is an assumption.

Response to the reviewer: We thank the reviewer for the suggestion. A change has been made in the manuscript

Changes in the manuscript: Edits reflected in Line 590 of ‘Revised Manuscript with Track Changes’

16. Lines 501-503. Following STROBE is not a strength of the study, but of the manuscript writing. Please remove.

Response to the reviewer: We thank the reviewer for the suggestion. The statement is deleted.

Changes in the manuscript: Edits reflected in Lines 604-606 of ‘Revised Manuscript with Track Changes’

17. Line 511. Briefly expand on this, and how it limits your ability to recommend changes/improvements. What information would be needed to inform these gaps and how would you propose to gather it?

Response to the reviewer: We thank the reviewer for the suggestion. Further information could be gathered using qualitative techniques, bottle neck analysis and inclusion of the entire lab network in a similar analysis.

Changes in the manuscript: Edits reflected in Lines 620-623 of ‘Revised Manuscript with Track Changes’

18. Lines 515-519. Here you have given specific examples of things to improve; however, there were no results proving that these were the specific issues to be addressed. Please revise.

Response to the reviewer: We thank the reviewer for the suggestion. We have rephrased this section.

19. Line 529. ‘Approximately’ is a preferred word to ‘about’. This sentence could be more concise.

Response to the reviewer: We thank the reviewer for the suggestion. A change has been made in the manuscript 

Changes in the manuscript: Edits reflected in Line 639 of ‘Revised Manuscript with Track Changes’

20. Line 531. ‘gaps’ should be expanded here – delays? not reaching the reference laboratory? not completing the algorithm?

Response to the reviewer: We thank the reviewer for the suggestion. Gaps have been expanded.

Changes in the manuscript: Edits reflected in Line 646-650 of ‘Revised Manuscript with Track Changes’

21. Lines 532-534. This should be included in the acknowledgements section rather than conclusions.

Response to the reviewer: We thank the reviewer for the suggestion. We beg to differ as we are not acknowledging the RNTCP staff for supporting our research but are just appreciating their routine performance based on the study findings. 

Changes in the manuscript: None

Tables:

Please be consistent across tables with capitalization and variable names. e.g. Table 2 non-key population; Table 4 Not Key population. Similarly, use consistent term for unavailable data for each variable. ‘Missing’ or ‘Not recorded’ or ‘Not available’. Also, with the site and laboratory.

Response to the reviewer: We thank the reviewer for the suggestion. 

Changes in the manuscript: A change has been made in the table of ‘Revised Manuscript with Track Changes’.

Table 2.

1. Under key population: reword to ‘persons living with HIV’

Response to the reviewer: Thanks for the reviewing and suggesting. As mentioned earlier, PLHIV refers to “people living with HIV” as per WHO and National programme documents and therefore we would retain this expression.

Changes in the manuscript: None

2. Under key population: Footnote should indicate what population is represented in ‘Others’

Response to the reviewer: We thank the reviewer for the suggestion Footnote added under Table 2.

Changes in the manuscript: Footnote added under Table 2.

3. For Specimen condition of receipt at Xpert lab – these categories only apply to sputum specimens? The table should reflect this.

Response to the reviewer: We thank the reviewer for the suggestion. A change has been made in the manuscript Changes in the manuscript: Footnote added under Table 2.

Table 3.

1. The last line FL-LPA result was resistant – to any first line drug?

Response to the reviewer: We thank the reviewer for the suggestion. A change has been made in the manuscript

Changes in the manuscript: Changes reflected in the last line of table 3.

Table 4.

1. Male reference is not indicated in aRR column

Response to the reviewer: We thank the reviewer for the suggestion. A change has been made in the manuscript 

2. There is no footnote for the abbreviation PLHIV

Response to the reviewer: We thank the reviewer for the suggestion. A change has been made in the manuscript 3. The N and (%) columns presumably refer the those that did not reach the laboratory. Please clarify this in the column header.

Response to the reviewer: We thank the reviewer for the suggestion. N explained in the footer.

Table 5.

1. As in Table 4, please be specific for the column header N (%)

Response to the reviewer: We thank the reviewer for the suggestion. N explained in the footer.

2. Age 0-14 is not an appropriate reference given that the N = 0 for non-completion.

Response to the reviewer: Thanks for the reviewing and suggesting. The Reference here is actually 60 and above and the changes are done accordingly.

Changes to the manuscript: We have made changes to the table.

3. Please indicate what the NA values represent.

Response to the reviewer: We thank the reviewer for the suggestion. A footnote is added.

4. Under Specimen type, remove the word sample for Extra-pulmonary

Response to the reviewer: We thank the reviewer for the suggestion. 

Changes to the manuscript: A change has been made in the table 

Figures:

Figure 2.

1. The denominator for culture done does not appear to add up. 39+21+43+4 = 107

Response to the reviewer: Thanks for the reviewing and suggesting. 

Corrections made in Figure 2. The denominator for culture done is 107. The edits are reflected in the figure 2

2. For the 3 Xpert RIF-S samples that were then RIF-R as the reference laboratory what was the pattern of INH resistant? This could be included as a footnote.

Response to the reviewer: Thanks for the reviewing and suggesting. 

Changes to the manuscript: A footnote depicting the same is reflected.

Reviewer #2: Overall comments: Thank you for the opportunity to review this manuscript. It is a well written manuscript that addresses a topic that is of fundamental importance to TB care in India.

Abstract: No major comments

Introduction:

1. Line 106: Is this prevalence of MDR-TB among all cases, if so I think specify.

Response to the reviewer: We thank the reviewer for the suggestion. The prevalence of MDR-TB is 6.2% among all cases.

Changes in the manuscript: Edits reflected in Lines 110 of ‘Revised Manuscript with Track Changes’

2. Line 108: Spell out XDR TB at first use

Response to the reviewer: We thank the reviewer for the suggestion. XDR expanded.

Changes in the manuscript: Edits reflected in Lines 111-113 of ‘Revised Manuscript with Track Changes’

3. Line 112: I think it would be good to have a brief explanation of what PMDT is.

Response to the reviewer: We thank the reviewer for the suggestion. PMDT provides guidelines for the integration of management of DR-TB with the existing National TB programme activities

Changes in the manuscript: Edits reflected in Lines 116-120 of ‘Revised Manuscript with Track Changes’

4. Line 116-117: is this statistic of 29% from India? Pls kindly clarify so that the context is clear.

Response to the reviewer: We thank the reviewer for the suggestion. This statistic is from India.

Changes in the manuscript: Edits reflected in Line 122-123 of ‘Revised Manuscript with Track Changes’

5. Line 126: You talk about non diagnosis as being one of the reasons for low treatment success but treatment success is really only measured for diagnosed cases, pls clarify.

Response to the reviewer: We thank the reviewer for the suggestion. Non-diagnosis has been removed.

Changes in the manuscript: Edits reflected in Line 133-135 of ‘Revised Manuscript with Track Changes’

6. Paragraph starting at line 130: I would have liked to know about more about the rollout of Xpert in India, can you provide a couple more sentences about this including the dates of rollout and how quickly it happened?

Response to the reviewer: We thank the reviewer for the suggestion. The feasibility study and scaling up details added. 

Changes in the manuscript: Edits reflected in Lines 171-176 of ‘Revised Manuscript with Track Changes’

7. Line 133: When did the policy of universal DST start? A date would be helpful.

Response to the reviewer: We thank the reviewer for the suggestion. Start date of universal DST has been added. 

Changes in the manuscript: Edits reflected in Line 176-179 of ‘Revised Manuscript with Track Changes’

Methods:

1. Line 183: I think this sentence about the population size needs a reference.

Response to the reviewer: We thank the reviewer for the suggestion. The reference added.

2. You mention the human carriers or couriers in lines 199-200 and then again in lines 208-209 which I think is repetitious.

Response to the reviewer: We thank the reviewer for the suggestion. The second reference to couriers and human carriers has been removed.

Changes in the manuscript: Edits made in line 261-262 of ‘Revised Manuscript with Track Changes’

3. Line 216-217: Is this second sample also tested using Xpert or LPA?

Response to the reviewer: We thank the reviewer for the query. If rifampicin resistance is detected in a new patient, the second sample is used at the Xpert laboratory for a repeat Xpert® MTB/RIF test to confirm rifampicin resistance.

Changes in the manuscript: None

4. Line 223: I think it would be better to say “If additional resistance” rather than “If resistant” as I think this is what is meant, i.e. if there is additional resistance then a pre XDR or XDR regimen is started.

Response to the reviewer: We thank the reviewer for the suggestion. A change has been made in the manuscript 

Changes in the manuscript: Edits made in line 279 of ‘Revised Manuscript with Track Changes’

5. I was wondering why a mobile phone number was used as the second method of identifying people rather than the name-age-sex combination which may be more unique. How well does a mobile phone number identify the user? Has this method been previously validated for matching people in population based studies? I think this needs further discussion and justification.

Response to the reviewer: We thank the reviewer for the query and the suggestion. The primary tracking was done with NIKSHAY Id. The mobile number was not used for identifying people. The mobile no. was used to match the records from two different lab registers. Name-age-sex being non-numerical variable may not provide a better match.

Changes in the manuscript: No changes done

6. In Table 1 I think some additional clarity is needed, i.e. for the third to fifth bullet points what is the resistance or sensitivity to? I think some additional detail is needed here. It should also be clear why completion of the diagnostic algorithm was constructed the way it was including having the denominator start at the reference laboratory as the diagnostic algorithm actually seems to start before then, i.e. in the Xpert laboratory.

Response to the reviewer: We thank the reviewer for the query and the suggestion. We have reviewed this and have made small edits to improve clarity. In the 3rd to 5th bullets, resistance on SL-LPA/FL-LPA refers to resistance to the SL-LPA/FL-LPA class of drugs. Since LPA reports resistance to a class of drugs and not to specific drugs per se. There were two key steps to the cascade we were assessing: 1) samples not-reaching the reference laboratory 2) not completing the algorithm after reaching the reference laboratory. The issues related to each of these steps are different and hence, we kept the analysis separate and used different denominators (total number of patients when assessing non-reaching and only those who reached for assessing non-completion). Using the total number as denominator for assessing non-completion would have mixed up issues of both non-reaching and non-completion, thus creating confusion. I hope this clarifies.

Changes in the manuscript: No changes done.

Results:

Overall the results section was well constructed and clear. 

1. My main comment relates to the numbers and Figures 2 and 3 and the definition of having completed the diagnostic algorithm. For Figure 2 I am not 100% sure how you got the figure of 103 in the culture done box, should this be 107 (i.e. 4 plus 43 plus 21 plus 39)? If I follow the lines on all of the boxes that lead the culture done box I get 107 instead of 103. 

Response to the reviewer: We thank the reviewer for the comment. Yes, there was an error in the calculation in figure 2. The relevant corrections have been made in figure 2 and some details are added in figure 3. The numbers/numerators in the greyed-out boxes add up to total completing the diagnostic algorithm in both figure 2 and 3.

2. And I wondered why the people who are susceptible or who had culture are the only ones who are deemed eligible to have completed the algorithm? If there is resistance on FL LPA and then that person goes on to have the appropriate tests, they have also completed the algorithm haven’t they? 

Response to reviewer: We thank the reviewer for the comment. We would like to clarify that those resistant on FL-LPA are considered to have completed the diagnostic algorithm if the appropriate tests are done. Those resistant on FL-LPA are required to undergo SL-LPA and then culture DST depending on the results of the SL-LPA. This is reflected in the following categories of patients completing diagnostic algorithm (Table 1):

• Rifampicin sensitive on Xpert® MTB/Rif and resistance to Isoniazid and/or Rifampicin on FL-LPA and sensitive to second-line drugs on SL-LPA

• Rifampicin sensitive on Xpert® MTB/Rif and resistance to Isoniazid and/or Rifampicin on FL-LPA and resistant on SL-LPA and culture done

3. In Figure 3 should there be a line from the box results not available to the box culture done so that the total is 26 and not 24?

Response to reviewer: We thank the reviewer for the suggestion. Samples in whom no results were available on SL-LPA could be invalid results and they should ideally be subjected to culture and DST to complete the algorithm (Refer to Category “Culture done on specimens with indeterminate or no results on either FL-LPA or SL-LPA” in Table 1) Therefore, the line linking “No results available’ to the ‘Culture done box’ has been drawn. We have also made the corrections to the total as 26. 

4. I also wondered if your denominators should really be 1590 and 64 rather than the denominators that you have as this is where the algorithm starts. 

Response to reviewer: We thank the reviewer for the comment. As explained earlier, there were two key steps to the cascade: 1) samples not-reaching the reference laboratory 2) not completing the algorithm after reaching the reference laboratory. The issues related to each of these steps are different and hence, we kept the analysis separate and used different denominators (total number of patients when assessing non-reaching and only those who reached for assessing non-completion). Using the total number as denominator for assessing non-completion would have mixed up issues of both non-reaching and non-completion, thus creating confusion. I hope this clarifies. 

5. For Figure 3 I also wondered if the people who completed the algorithm should be the 14 who had SL LPA and then any additional people who had culture when it was indicated. I think the 9 people who were FQ and SLI susceptible are include in the numerator of 17 but if you are resistant doesn’t it also mean that you have completed the algorithm?

Response to reviewer: We thank the reviewer for the comment. You are right that the 9 people who were FQ and SLI susceptible are included in the numerator of 17. We would like to clarify that detection of resistance on SL-LPA does not indicate completion, they would still need to undergo culture DST so that resistance patterns to individual drugs can be identified.

As shown in Fig 3 and definitions of Table 1, the following is the distribution of the 17 who were labelled as having completed the diagnostic algorithm:

• FQ and SLI sensitive box �9 individuals 

• Culture done when it was indicated (Rifampicin resistant on Xpert® MTB/Rif and resistant on SL-LPA, specimens with indeterminate or no results on either FL-LPA or SL-LPA) 8 individuals 

Discussion:

1. Line 461: I think you should reference the “previous studies” referred to here and as a general comment I think there could be more use of other studies in the Discussion section as it mainly focuses on the findings of the study rather than comparing and contrasting with other literature from India, the region or elsewhere. There is one study mentioned in lines 461-463 but it is not clear what date this was and it is a study on EPTB so may not be directly comparable to your overall sample as the majority of your sample were PTB (although admittedly it does seem that EPTB samples were less likely to be referred to the reference laboratory). 

Response to reviewer: We thank the reviewer for the comment. The one study referenced was not only EPTB but also had PTB samples. Since the integrated algorithm was introduced in India, this is the first assessment of the complete integrated diagnostic algorithm. Recommended reference standards are not available for the processes involved in the completion of the diagnostic algorithm, the time for taken for specimen collection, transportation and reporting of the results, except for turnaround time for testing. Hence, we could not compare with other studies.

2. I think it could be emphasized a bit more the loss of specimens going from the Xpert lab to the reference lab and the implications of this. I think you could also emphasise the losses for the RR cases as well as these are the very cases that you would want to know have completed the diagnostic algorithm.

Response to reviewer: We thank the reviewer for the comment. We have made changes to the manuscript. 

Changes in the manuscript: Edits made in lines 551-561 of ‘Revised Manuscript with Track Changes’

3. Under the section on Strengths you talk about sensitivity and specificity of individual tests but you did not do this so I would recommend leaving this out.

Response to reviewer: Thank for the comments. The line on sensitivity and specificity has been removed.

Changes in manuscript: Edits made in line numbers 601-602 of ‘Revised Manuscript with Track Changes’

Reviewer #3: Comments:

The subject of the manuscript has merit and describes important findings related to Drugs resistant TB diagnostic algorithm under routine programme settings in India. The authors may address following queries to strengthen the manuscript.

Major concerns:

Introduction- Introduction may need restructuring.

1. The first paragraph seems general and it discusses about prevention, diagnosis and treatment while the manuscript is only about diagnostic cascade.

Response to reviewer: We thank the reviewer for the comment. This is to provide general information about the TB, globally and in India to all those who read, and more so to the general readership of PLoS ONE who may need more context to gain a comprehensive understanding of the programme in India.

Changes in the manuscript: None

2. Line 124-128: It describes about treatment success and its linkage to delay in diagnosis. This section could be mentioned in the discussion.

Response to the reviewer: We thank the reviewer for the comment. We feel this is best placed in the introduction because we want to emphasize upfront the importance of early and accurate diagnosis for successful treatment of TB, especially DR-TB.

Changes in the manuscript: None

Methods:

1. Line 241: How were these ten districts selected for the study? Please provide some information.

Response to the reviewer: We thank the reviewer for the query. The districts were selected conveniently based on feasibility of data collection (and all laboratories within a district were included).

2. The study mentions about gap in UDST, however the last steps of the algorithm is considered as Culture done. I wonder if it must end at DST level (for how many had DST was done). Though the operational definition mentions the algorithm finishes at culture done, the authors may want to describe about this concern in the manuscript.

Response to the reviewer: We thank the reviewer for this suggestion. We agree that ideally the algorithm should have ended with samples in which the results of culture DST were available. Since culture DST takes time, we might not have been able to extract results of culture and DST within the timeframe of the study. Hence, we operationally defined completion if the samples were subjected to culture and DST.

Changes in the manuscript: Edits made in lines 179-180 of ‘Revised Manuscript with Track Changes’

Results:

1. Line 347-348: Please check if the total eligible for culture was 107 instead of 103.

Response to the reviewer: Thanks for the observation. The relevant corrections have been made in figure 2

Discussion:

1. Line 448-449: Since the integrated DR-TB diagnostic algorithm is specific for India, you may tone down the first statement.

Response to the reviewer: We thank the reviewer for the comment. We have made changes in the manuscript. 

Changes in the manuscript: correction done in line 532 of ‘Revised Manuscript with Track Changes’

2. Line 457-459 seems a repeat of the results, please review.

Response to the reviewer: We thank the reviewer for the suggestion. We have tried to summarise key results in the discussion section for ease of reading and comprehension. 

3. Line 489: when the authors mention about the delays, they may consider that availability of dates for SL-LPA was quite low. The claims could be toned down.

Response to the reviewer: We thank the reviewer for the suggestion. We have rephrased this statement to “This may explain why delays with SL-LPA were three times more when compared to FL-LPA, though valid dates of receipt and reporting of SL-LPA samples were found for few samples.”

Changes in the manuscript: Edits reflected in lines 590-592 of ‘Revised Manuscript with Track Changes’

Minor concerns:

1. Please update the references. For example, Line 105 must include the recent literature (Global TB Report 2019).

Response to the reviewer: We thank the reviewer for the suggestion. 

Changes in the manuscript: Edits reflected in lines 105 & 108 of ‘Revised Manuscript with Track Changes’

2. Line 167: no systematic assessment. do we mean. in India? If yes, you may want to mention this.

Response to the reviewer: We thank the reviewer for the comment. We have made this change.

Changes in the manuscript: Edits reflected in line 210 of ‘Revised Manuscript with Track Changes’

3. Line 199: change ‘…were transported.’ to ‘...are transported.’

Response to the reviewer: We agree to this suggestion and have made this change.

Changes in the manuscript: Edits reflected in line 250 of ‘Revised Manuscript with Track Changes’

4. Line 200-202: This should be mentioned under the Programme implementation part of the Methods section.

Response to the reviewer: We thank the reviewer for this suggestion. However, the support provided by the NGO is specific to Bengaluru city and adjacent districts and not a routine feature in the programme. Therefore, it has been mentioned in the “Specific Setting” section.

5. Line 204-237: The integrated diagnostic algorithm: Can the authors summarize the section, as the same is described in the Figure 1.

Response to the reviewer: We thank the reviewer for this suggestion. There have been comments from other reviewers requesting additional details in this write up, so that readers who are not familiar with the algorithms followed in India may be able to understand it better. Therefore, we would like to retain this.

6. Line 251- double entry and validation, wherever possible: please explain where it was carried out and where it was not possible.

Response to the reviewer: We thank the reviewer for the comment. Double entry and validation were done for the data entered from the laboratory registers of Xpert lab and not for the data entered from the lab registers of reference laboratories.

Changes in the manuscript: Edits reflected in lines 313 - 315 of ‘Revised Manuscript with Track Changes’

7. Line 309: Were the key population mutually exclusive group. If someone was urban slum dweller and PLHIV, in which category they were considered?

Response to the reviewer: We agree that these categories may not be mutually exclusive, but the registers maintained at the laboratories allow entry of only one option and hence whatever was recorded was considered.

8. Table 2: It is good to mention in the title of the table that 13 Xpert laboratories were included in the study

Response to the reviewer: We agree to this suggestion and have made this change in the Title of Table 2.

Changes in the manuscript: Edits reflected in line 405 of ‘Revised Manuscript with Track Changes’

9. Line 461: Please add reference to the statement.

Response to the reviewer: Thanks for the suggestions. I have added three references.

Changes in the manuscript: Edits reflected in line 555 of ‘Revised Manuscript with Track Changes’

10. Line 474: In Methods it was mentioned that the NGO was working in all selected districts of study, please review and change the statement.

Response to the reviewer: We would like to clarify that the NGO was working only in few districts. 

Please refer to the Revised manuscript (Line 220-222) under the specific settings section. “During the study period, a Non-Governmental Organization (NGO) was assisting in transportation of the samples between the laboratories in districts around Bengaluru city….” 

Reviewer #4: I wish to congratulate the authors on this very clear and helpful paper. It is a transparent analysis of an operational challenge in TB control, which will be of benefit to others working in the same field. I however do have a small number of concerns that I would suggest the authors address, before recommending this manuscript for publication:

1. This study on the performance of the UDST in Karnataka was conducted only a few months after its implementation (data from July-August, for a system implemented in April). Could the authors comment on whether the results are likely to be affected by the study being conducted in this early phase? Could there be "teething troubles" with the UDST, or conversely could there be an ambitious start, which then deteriorates over time?

Response to the reviewer: We thank you for this suggestion. The guidelines for the implementation of UDST were given by the Government of India in 2017, while Karnataka had all its districts covered under UDST by April 2018. While the overall results are encouraging, there are some gaps that are worrisome. We are not able to comment on whether this could be attributed to teething troubles or ambitious start. Such an assertion would probably best be made after conducting a detailed situational analysis as mentioned in the conclusion section of the manuscript. 

Changes in the manuscript: Edits reflected in line 176-179 of ‘Revised Manuscript with Track Changes’

2. In the same vein: the study was conducted only over 2 months (July-August) - do the authors anticipate any seasonality in the performance of the UDST?

Response to the reviewer: To the best of our knowledge, it is unlikely that seasonality has played a role in the performance of UDST. However, to be able to validate this assumption, we would need to look at the indicators for at least a year, which was outside the scope of this study.

3. The study hinges to a large extent on the matching algorithm that was used between database 1 and database 2, relying on the Nikshay number, phone number, and a name/age/sex match. This is a commendable effort, but it is not without risk. I would recommend that the authors report on how well the matching worked (which proportion was matched on Nikshay, which on phone number, etc.); possibly as supplemental material. Additionally, if I understand well, all entries in database 2 should have a corresponding match in database 1 (as no patients would end up directly at the reference lab) - any "unmatched" individuals in database 2 would therefore represent a measure of how many incorrect matchings resulted from the algorithm, and this may be worth reporting on.

Response to the reviewer: Thanks for this insightful comment. Unfortunately, we did not keep a record of the number of matches obtained by each of the identifiers (such as Nikshay ID, phone number and name-age-sex). Hence, we will not be able to provide this information. However, we would like to clarify that unmatched individuals in database 2 do not necessarily represent a measure of incorrect matchings because, the reference laboratory also received samples from other districts (not included in the study) and, sometimes also received direct samples from private providers without routing through the Xpert laboratories located in the districts. 

4. Line 290-291: I am not entirely clear on the "exploratory approach" applied, and/or why *all* factors were included in the adjusted analysis.

Response to the reviewer: Linked to 1st Reviewer comment on use of gender as a factor

5. While I appreciate the various ethics reviews done, I would suggest to expand on how patient confidentiality was protected, given the extensive use of phone numbers and names in this study.

Response to the reviewer: We highly appreciate the concern regarding patient confidentiality. As noted, certain identifiers had to be extracted from records to enable linkage across databases. The data was entered on to a restricted access password protected electronic data capture system (EpiData) which was installed on secure desktop systems in the laboratories. Only the investigators had access to the data with identifiers required for linking databases. Once linked, the final dataset was stripped of all identifiers prior to further analysis. 

Changes in the manuscript: Edits reflected in line nos. 316-318 of ‘Revised Manuscript with Track Changes’

6. Line 348-350 ("Thus, a total of 1106 (95%) of rifampicin-sensitive TB patients were considered to have completed the diagnostic algorithm.") is perhaps phrased a bit too optimistically, given that a large proportion of RS TB samples did not even show up at the reference lab. I would suggest correcting to "(...) of rifampicin-sensitive TB patients whose samples were successfully received at the reference laboratory were considered to have completed the diagnostic algorithm." Same comment for line 363-364.

Response to the reviewer: We agree and the statement has been rephrased.

Changes in the manuscript: Edits reflected in line 420-421 of ‘Revised Manuscript with Track Changes’

7. On a minor note: one of the percentages in table 3 is incorrect (9.1 should read 91.4).

Response to the reviewer: Thank you for bringing this to our notice. We have made the correction in Table 3

Changes in the manuscript: Edits reflected in Table 3 of ‘Revised Manuscript with Track Changes’

8. In the methods section the authors refer to "selected districts” and in the limitations to "selected laboratories". Could they clarify the selection process, and which criteria were used?

Response to the reviewer: The districts selected conveniently based on feasibility of data collection, but all the Xpert/CBNAAT sites within the selected districts were included.

Changes in the manuscript: Edits reflected in line 614-615 of ‘Revised Manuscript with Track Changes’

---

## [Decision Letter · Decision Letter 1]

8 Oct 2020

PONE-D-19-25977R1

Implementation of the new integrated algorithm for diagnosis of drug resistant tuberculosis in Karnataka State, India: How well are we doing?

PLOS ONE

Dear Dr. Shankar S,

Thank you for submitting your manuscript to PLOS ONE. Its was re-reviewed by one of the initial reviewers who reiterated his/her criticisms and further revision is required based on the recommendations made by the reviewer.

We look forward to receiving your revised manuscript.

Kind regards,

Igor Mokrousov, Ph.D., D.Sc.

Academic Editor

PLOS ONE

Reviewers' comments:

Reviewer's Responses to Questions

**Comments to the Author**

1. If the authors have adequately addressed your comments raised in a previous round of review and you feel that this manuscript is now acceptable for publication, you may indicate that here to bypass the “Comments to the Author” section, enter your conflict of interest statement in the “Confidential to Editor” section, and submit your "Accept" recommendation.

Reviewer #1: (No Response)

2. Is the manuscript technically sound, and do the data support the conclusions?

Reviewer #1: Partly

3. Has the statistical analysis been performed appropriately and rigorously? 

Reviewer #1: I Don't Know

4. Have the authors made all data underlying the findings in their manuscript fully available?

Reviewer #1: Yes

5. Is the manuscript presented in an intelligible fashion and written in standard English?

Reviewer #1: Yes

6. Review Comments to the Author

Reviewer #1: Comments: Thank you for the opportunity to review this revised manuscript. The authors have made changes to clarify statements in the paper and fix errors as recommended by the reviewers. However, the authors have declined to address issues raised by reviewers that would allow the reader to properly assess their findings and that would improve the manuscript overall, particularly in the methods, results and discussion.

Major issues:

1. The rationale for including patient-level variables in the model was not included in the methods section as requested. As the manuscript is currently written, it remains unclear to the reader why these variables would confound the association between specimen-specific variables and ‘not reaching’. If they are being used as proxies for something related to the specimen then this information should be included in the methods section. If for example, the authors in their response to reviewers have identified that specimens are inherently different when collected from children and this may be associated with ‘not reaching’, then perhaps use broader age categories that have only 2 or 3 levels (children, adult, older adult age groups).

Same for key population. These populations are not mutually exclusive and treating them as discrete populations is likely to create noise in the model and not provide a meaningful result. I suspect there are individuals that are HIV positive, reside in a slum and use tobacco. It is recommended that this variable be aggregated to a simple Yes/No of belonging to a key population. Please include in the methods the justification for including patient variables. It is understandable why key population would be associated with being tested, but once a specimen is collected the reason the specimen wouldn’t make it through the algorithm is not clearly outlined in the manuscript with respect to key population or other patient-centered variables. Please support where possible with references and explanation. While this may be the first such TB study in India there are many studies of other similar algorithms, diseases and/or populations from which to draw information from.

Without clear rationale it will be assumed that this was the data available to the authors and so it was included in the model without any consideration of how to best incorporate it (appropriate levels within variables) and why it should be.

2. It was requested that model fitting information be reported. The authors declined to do this.

Key practices of model building and validation, and reporting of results are required for reviewers and readers to assess the findings. It will be necessary to include:

• Standard (ANOVA table) model output, either in the body of the paper or as supplementary materials. All statistics packages produces this as output. A table for each of the final models showing the estimate, SE, df, test statistic and p value for each term in the model.

• Tests of model assumptions (e.g. tested for overdispersion and found…)

3. Typically, calculation of RR from a Poisson model should use robust error variance as the standard confidence intervals are not valid. Based on the information available regarding the models it is not clear if this was done. Please ensure the appropriate measure of variation was used and detail what was done in the methods section.

4. The authors should also address limitations of the models in the Discussion (e.g., no examination of interaction terms, possibility of low statistical power for some tests)

Minor:

1. It was requested that the authors indicate if multiple specimens were included for individuals. Information regarding this was provided in the response but not incorporated into the manuscript. Please include this in the methods section to provide the reader with this pertinent info.

2. The number not reaching was not clear based on the figures and clarification of why this is was provided in the response to reviewers but not incorporated into the manuscript. A separate figure or inclusion of another cascade in either figure 2 or 3 would be helpful to make this clear to readers. In addition details regarding these 6 negative/indeterminate should be included prior to the “Overall, out of the total 1660 samples,” otherwise it is not clear why the numbers from the RS and RR do not add to the overall numbers.

7. PLOS authors have the option to publish the peer review history of their article (what does this mean?). If published, this will include your full peer review and any attached files.

Reviewer #1: No

---

## [Author Response · Author response to Decision Letter 1]

22 Nov 2020

Review Comments to the Author

Reviewer #1: Comments: Thank you for the opportunity to review this revised manuscript. The authors have made changes to clarify statements in the paper and fix errors as recommended by the reviewers. However, the authors have declined to address issues raised by reviewers that would allow the reader to properly assess their findings and that would improve the manuscript overall, particularly in the methods, results and discussion.

Major issues:

1. The rationale for including patient-level variables in the model was not included in the methods section as requested. As the manuscript is currently written, it remains unclear to the reader why these variables would confound the association between specimen-specific variables and ‘not reaching’. If they are being used as proxies for something related to the specimen then this information should be included in the methods section. If for example, the authors in their response to reviewers have identified that specimens are inherently different when collected from children and this may be associated with ‘not reaching’, then perhaps use broader age categories that have only 2 or 3 levels (children, adult, older adult age groups).

Ans. We thank the reviewer for the comment. We accept that it is not justifiable to include patient level variables and hence have decided to do away with inclusion of the patient level variables both for non- reach and non-completion tables.

Same for key population. These populations are not mutually exclusive and treating them as discrete populations is likely to create noise in the model and not provide a meaningful result. I suspect there are individuals that are HIV positive, reside in a slum and use tobacco. It is recommended that this variable be aggregated to a simple Yes/No of belonging to a key population. Please include in the methods the justification for including patient variables. It is understandable why key population would be associated with being tested, but once a specimen is collected the reason the specimen wouldn’t make it through the algorithm is not clearly outlined in the manuscript with respect to key population or other patient-centered variables. Please support where possible with references and explanation. While this may be the first such TB study in India there are many studies of other similar algorithms, diseases and/or populations from which to draw information from.

Without clear rationale it will be assumed that this was the data available to the authors and so it was included in the model without any consideration of how to best incorporate it (appropriate levels within variables) and why it should be.

Ans. We accept that key population too constitute patient level data, hence we have decided to exclude this variable too from both the tables of non-reach and non-completion.

2. It was requested that model fitting information be reported. The authors declined to do this.

Key practices of model building and validation, and reporting of results are required for reviewers and readers to assess the findings. It will be necessary to include:

• Standard (ANOVA table) model output, either in the body of the paper or as supplementary materials. All statistics packages produces this as output. A table for each of the final models showing the estimate, SE, df, test statistic and p value for each term in the model.

• Tests of model assumptions (e.g. tested for overdispersion and found…)

Ans. The final model output table will be attached as a supplementary file as suggested.

3. Typically, calculation of RR from a Poisson model should use robust error variance as the standard confidence intervals are not valid. Based on the information available regarding the models it is not clear if this was done. Please ensure the appropriate measure of variation was used and detail what was done in the methods section.

Ans. Yes robust error variance was done. We will attach the final model output sheet and table depicting the same as a supplementary file.

4. The authors should also address limitations of the models in the Discussion (e.g., no examination of interaction terms, possibility of low statistical power for some tests)

Ans. We have not examined any interactions, due to the fewer variables we had and the low statistical power. The same will be mentioned in line nos. 498,499.

Minor:

1. It was requested that the authors indicate if multiple specimens were included for individuals. Information regarding this was provided in the response but not incorporated into the manuscript. Please include this in the methods section to provide the reader with this pertinent info.

Ans. Thanks for this valuable observation. We have mentioned about the same in line number 272 and 273 of the revised manuscript.

2. The number not reaching was not clear based on the figures and clarification of why this is was provided in the response to reviewers but not incorporated into the manuscript. A separate figure or inclusion of another cascade in either figure 2 or 3 would be helpful to make this clear to readers. In addition details regarding these 6 negative/indeterminate should be included prior to the “Overall, out of the total 1660 samples,” otherwise it is not clear why the numbers from the RS and RR do not add to the overall numbers.

Ans. Thanks for the suggestion, the description about the 6-rifampicin resistance reports not available/indeterminate has been mentioned at two appropriate paragraphs, one in line nos. 331,332 and another in line nos. 344,345.

---

## [Decision Letter · Decision Letter 2]

17 Dec 2020

Implementation of the new integrated algorithm for diagnosis of drug resistant tuberculosis in Karnataka State, India: How well are we doing?

PONE-D-19-25977R2

Dear Dr. Shankar S,

We’re pleased to inform you that your manuscript has been judged scientifically suitable for publication and will be formally accepted for publication once it meets all outstanding technical requirements.

Kind regards,

Igor Mokrousov, Ph.D., D.Sc.

Academic Editor

PLOS ONE

Additional Editor Comments (optional):

Reviewers' comments:

Reviewer's Responses to Questions

**Comments to the Author**

1. If the authors have adequately addressed your comments raised in a previous round of review and you feel that this manuscript is now acceptable for publication, you may indicate that here to bypass the “Comments to the Author” section, enter your conflict of interest statement in the “Confidential to Editor” section, and submit your "Accept" recommendation.

Reviewer #1: All comments have been addressed

2. Is the manuscript technically sound, and do the data support the conclusions?

Reviewer #1: Yes

3. Has the statistical analysis been performed appropriately and rigorously? 

Reviewer #1: Yes

4. Have the authors made all data underlying the findings in their manuscript fully available?

Reviewer #1: Yes

5. Is the manuscript presented in an intelligible fashion and written in standard English?

Reviewer #1: Yes

6. Review Comments to the Author

Reviewer #1: (No Response)

7. PLOS authors have the option to publish the peer review history of their article (what does this mean?). If published, this will include your full peer review and any attached files.

Reviewer #1: No

---

## [Editor Report · Acceptance letter]

23 Dec 2020

PONE-D-19-25977R2 

Implementation of the new integrated algorithm for diagnosis of drug-resistant tuberculosis in Karnataka State, India: How well are we doing? 

Dear Dr. Shankar S:

I'm pleased to inform you that your manuscript has been deemed suitable for publication in PLOS ONE. Congratulations! Your manuscript is now with our production department. 

Kind regards, 

on behalf of

Dr Igor Mokrousov 

Academic Editor

PLOS ONE